

# Calcium content and high calcium adaptation of plants in karst areas of southwestern Hunan, China

Xiaocong Wei[1], Xiangwen Deng[1,2*], Wenhua Xiang[1,2] Pifeng Lei[1,2], Shuai Ouyang[1,2], Hongfang Wen[1], Liang Chen[1,2]

[1] Faculty of Life Science and Technology, Central South University of Forestry and Technology, Changsha 410004, Hunan Province, China

[2] Huitong National Field Station for Scientific Observation and Research of Chinese Fir Plantation Ecosystem in Hunan Province, Huitong 438107, China

*Correspondence to*: Xiangwen Deng, Email: dxwfree@126.com, Tel.: +86 0731 85623483

**Abstract.** Rocky desertification is a major ecological problem of land degradation in karst areas. Its high soil calcium ($Ca^{2+}$) content has become an important environmental factor which can affect the restoration of vegetation in such rocky desertification areas. Consequently, the screening of plant species, which can adapt to soils high $Ca^{2+}$ environment, is a critical step for vegetation restoration. However, the $Ca^{2+}$ dynamics of plants and soil are not well understood. In this study, three different grades of rocky desertification samples areas (LRD, light rocky desertification; MRD, moderate rocky desertification; IRD, intense rocky desertification) were selected in karst areas of southwestern Hunan, China. Each grade of these samples areas had 3 sample plots in different slop positions, each of which had 4 small quadrats (1 in rocky side areas, 3 in non-rocky side areas). We measured the $Ca^{2+}$ content of leaves, branches and roots from 41 plant species, as well as soil total $Ca^{2+}$ (TCa) and exchange $Ca^{2+}$ (ECa) at depths of 0–15, 15–30 and 30–45 cm under each small quadrat. The results showed that the soil $Ca^{2+}$ content in rocky side areas was significantly higher than that in non-rocky side areas ($p<0.05$). The mean soil TCa and ECa content increased gradually along with the grade of rocky desertification, in the order IRD > MRD > LRD. For all plant functional groups, the plant $Ca^{2+}$ content of aboveground parts was significantly higher than that of the underground parts ($p<0.05$). The soil ECa content had significant effects on plant $Ca^{2+}$ content of the underground parts, but had no significant effects on plant $Ca^{2+}$ content of the aboveground parts. According to the differences in $Ca^{2+}$ content between the aboveground and underground parts of 17 dominant species (important value, *IV*>1) and their correlations with soil ECa

content, these 17 species can be divided into three categories: Ca-indifferent plants, high-Ca plants and low-Ca plants. Our results can provide a vital theoretical basis and practical guide for vegetation restoration and ecosystem reconstruction in rocky desertification areas.

**Keywords:** Rocky desertification; High calcium adaptation; Plant functional groups; Plant Ca content; Soil Ca content.

## 1 Introduction

Karst is a kind of typical calcium (Ca)-rich environment and a unique ecological environment system. This type of ecosystem is widely distributed, accounting for 12% of the world's total land area (Zeng et al., 2007; Zhou et al., 2009; Luo et al., 2012). Karst landforms in China are mainly distributed in southwestern areas. Of these, the severity of rocky desertification in Hunan Province was ranked fourth (Li et al., 2016). Rocky desertification is an extreme form of land degradation in karst areas, and has become a major social problem in terms of China's economic and social development (Sheng et al., 2015). The restoration and reconstruction of rocky desertification ecosystems has become the immediate focus of agro-forestry production environment improvements, regional economic development and helping to support people out of poverty (Jing et al., 2016). Soil high Ca content in rock desertification areas has become one of the most important environmental factors affecting the local plant physiological characteristics and distribution in these areas (Ji et al., 2009). From the origin of rocky desertification, the restoration of vegetation is key to the process of remediation (Wang et al., 2004). Consequently, the screening of plants which can grow successfully in high- calcium environments is an extremely critical step.

Role of $Ca^{2+}$ in plant physiology: $Ca^{2+}$ is one of the most essential nutrients needed for the regulation of plant growth and also plays a central role in helping plants overcome environmental stress (Hepler, 2005). A low cytosolic $Ca^{2+}$ concentration is crucial for appropriate cell signaling (Müller et al., 2015). At the same time, $Ca^{2+}$ is a versatile plant signal sensor under conditions of soil water stress (Hong-Bo and Ming, 2008). In addition, $Ca^{2+}$ as a second messenger in the process of cell signal transduction, which plays a key regulatory role in how plants respond to environmental changes (Poovaiah and Reddy, 1993; Batistič and Kudla, 2012). In the absence of nutrients (such as phosphorus), plants will inhibit the activity of nitrate reductase, thereby inhibiting the absorption of nitrate nitrogen, and ultimately inhibiting the absorption of $Ca^{2+}$ (Reuveni et al., 2000).





However, high calcium stress can exert influence over the photosynthetic and growth rate of plants (Ji et al., 2009; Hui et al., 2003). $Ca^{2+}$ and pectin in the cell walls of plants combine to form pectin calcium, which is a vital component of the cell wall (Kinzel, 1989). Ca also has the function of maintaining the structure and function of cell membranes, regulating the activity of biological enzymes, and maintaining the anion-cation balance in vacuoles (Marschner, 2011).

Mechanisms of plant defense to high soil $Ca^{2+}$ concentrations: $Ca^{2+}$ is an essential macronutrient, but low $Ca^{2+}$ concentrations must be maintained within the plant cytoplasm to avoid toxicity (Borer et al., 2012). Plants can be adapted to high salt environments by activating the $Ca^{2+}$ signal transduction pathway (Bressan et al., 1998). Drought is a common environmental stress factor in rocky desertification areas, and high temperatures enhance the degree of heat damage, causing oxidative damage to the cell membrane. However, if the $Ca^{2+}$ concentration of plants can be increased, this process can be

effectively inhibited, thereby preventing or reducing heat damage (Larkindale and Knight, 2002). A fine regulatory mechanism exists in the plant cell that cannot only rapidly increase the free $Ca^{2+}$ concentration of the cytoplasm to adapt to environmental changes, but also maintain a low Ca concentration to prevent harm caused by high Ca. This fine regulatory mechanism is mainly achieved by $Ca^{2+}$ channels, which play a key role in the $Ca^{2+}$ transport system in plants (Shang et al., 2003; Hetherington and Brownlee, 2004; Wang et al., 2005). The $Ca^{2+}$ transport system ($Ca^{2+}$ channel, $Ca^{2+}$ / $H^+$ reverse

transport carrier and $Ca^{2+}$-ATPase) plays an important role in the uptake, transport and distribution of Ca in plants (White and Broadley, 2003). The vacuoles may account for 95% of the plant cell volume and are able to store Ca within the cell. Thus, empty vacuoles represent an efficient means of Ca storage (Ranjev et al., 1993).

  Specific variability in plant $Ca^{2+}$ content and tolerance: The concentration of free $Ca^{2+}$ in vacuoles varies with plant species, cell type and environment, which may also affect the release of $Ca^{2+}$ in vacuoles (Peiter, 2011). The Ca content of plants

usually lies between 0.1 % and 5.0 %, and mostly exists in cell walls and vacuoles - in the form of pectin combination morphology and insoluble organic and inorganic Ca salts (Kinzel, 1989). Cytoplasmic $Ca^{2+}$ is mainly combined with proteins and other macromolecules; the concentration of free $Ca^{2+}$ is generally only 20–200 nmol $Ł^{-1}$ and is stored in cell gaps and organelles such as vacuoles, endoplasmic reticulum, mitochondria and chloroplasts (Wu, 2008). However, Excess free $Ca^{2+}$ in cytoplasm combines with phosphate to form a precipitate, which interferes with the physiological processes associated with

phosphorus metabolism, thus hindering normal signal transduction and causing significant detriment to plant growth (White and Broadley, 2003; Hirschi, 2004).

Ji et al. (2009) revealed that the mean soil ECa was 3.61 g kg$^{-1}$ in the Puding, Huajing, Libo and Luodian Counties of Guizhou Province, which is several times that of non-limestone areas in China. Zhang (2005) studied the growth habits of *Eurycorymbus caraleriel* and *Rhododendron decorum* under different concentrations of Ca$^{2+}$ and found that a high Ca$^{2+}$ concentration (50 mmol Ł$^{-1}$) could promote growth in *Eurycorymbus caraleriel*, but inhibit growth in *Rhododendron decorum*. These results indicate that there are differences in soil Ca content between different areas and that there are differences between calcareous and non-calcareous plants in terms of Ca absorption, transport and storage and other physiological processes. Collectively, these differences lead to different degrees of adaptability of plants to high Ca environments. However, to date, there is a scarcity of extensive research into the mechanisms by which plants adapt to high Ca conditions, particularly in karst areas.

In this study, we investigated plant Ca content, soil exchangeable Ca (ECa) and total Ca (TCa) contents on the rocky and non-rocky sides of three different grades of rocky desertification areas in southwestern China. Specifically, we hypothesized that the dynamics of Ca content in plants and soil would be significantly affected by the grade of rocky desertification. In order to test this hypothesis, we did the following: (i) to measure the soil ECa and TCa contents in rocky and non-rocky side areass; (ii) to investigate and compare the Ca content of aboveground and underground parts among of plants from different functional groups; and (iii) to reveal correlation between plant Ca content and soil ECa content.

## 2 Materials and methods

### 2.1 Site description

The study site was located in LijiaPing town of Shaoyang County, Hunan Province, China (latitude 27 ́0' N; longitude 113 ́36' E, elevation 400–585 m above sea level; see Fig. 1). This region experiences a humid mid-subtropical monsoon climate. Mean annual air temperature is 16.9 ℃, and maximum and minimum temperature is 41.0 ℃ and −10.1°C, respectively. Mean annual

precipitation is 1399 mm, mostly occurring between April and August, and the frost-free period is 288 days. The study site mainly consists of black and yellow lime soil, and vegetation is scarce. Groundwater level is low and groundwater storage is poor.

## 2.2 Data collection

Rocky desertification was graded by using the sum of four index scores: bedrock expose rate, vegetation type, vegetation coverage and soil thickness. These four main indices were quantified according to the State Forestry Administration of the People's Republic of China industrial standard 'LY/T 1840—2009' (China, 2009). Three 1 hm$^2$ sample areas were selected, which were each representative of the three different grades of rocky desertification: LRD, light rocky desertification; MRD, moderate rocky desertification; IRD, intense rocky desertification. Within each sample area, we recorded a range of

characteristics and data relating to the surrounding environment, including longitude, latitude, altitude, topography, vegetation type, degree of bare bedrock, and other conditions. We conducted a detailed survey of the three sample areas and collected samples in October 2016.

For each of the three sample areas, we assigned four 2×2 small quadrats in different slope positions (upper, middle, and lower slope). In total, we assigned 36 small quadrats for analysis. We chose to study the common plant species of the region,

and gathered plants using the whole plant harvest method. In each small quadrat, every kind of shrubs and herbs are collected. Shrubs were divided into three parts: branches, leaves and roots. Herbs were divided into two parts: aboveground and underground parts. Plant samples were taken back to the laboratory, rinsed with distilled water before being heated at 105 ℃ for 15 min to de-enzyme, dried to a constant weight at 80 ℃, crushed and passed through a 0.149 mm sieve, and bagged for later analysis. For the soil samples, we measured the TCa and ECa relating to the quadrat soil (top soil: 0-15 cm; middle soil:

15-30 cm; bottom soil, 30-45 cm). Finally, soil TCa, ECa content and plant Ca content, were measured using an Atomic Absorption Spectrophotometer (3510, Shanghai, China).

**Biogeosciences Discussions**

**2.3 Data analysis**

All plant species were divided into different functional groups: (1) nitrogen-fixing plants and non-nitrogen-fixing plants groups according to nitrogen-fixing function, (2) dicotyledons and monocotyledons groups according to system development type, (3) C3 and C4 plants groups according to photosynthetic pathway and (4) deciduous shrubs, evergreen shrubs, annual herbs and perennial herbs according to life form. 'Annual herbs' included both annual herbs and biennial herbs, while 'deciduous shrubs' included deciduous trees with a height less than 2 m or a ground diameter less than 3 cm. The aboveground part of plants included branches and leaves, while the underground part included roots. One-way analysis of variance (ANOVA), Two-way ANOVA and Pearson correlation analysis ($\alpha = 0.05$) were used to analyze the Ca content of soil and plants within and between different grades of rocky desertification. All statistical analyses were performed using R 3.3.3 (R Development Core Team, 2017).

**3 Results**

**3.1 The properties of soil in different grades of rocky desertification**

The mean TCa content was 2.40 g $kg^{-1}$ (range: 0.10–8.09 g $kg^{-1}$) while mean ECa content was 1.46 g $kg^{-1}$ (range: 0.02–3.92 g $kg^{-1}$). Differences between different sample points (non-rocky side and rocky side) were significant ($p<0.05$) for both TCa and ECa. Furthermore, mean soil TCa and ECa content were found to be the highest in areas of IRD, followed by MRD, followed by LRD. However, only the mean soil ECa content showed significant differences ($p<0.05$) across the three different grades of rocky desertification. Regarding the availability of Ca, the average was 59.75%, with MRD showing the highest content at 72.55%, followed by IRD at 58.98%, and LRD showing the lowest content at 47.72 % (Table. 2).

**3.2 The Ca content of plants**

**3.2.1 The Ca content of plant in different grades of rocky desertification areas**

The 41 plant species were collected from the three different grades of rocky desertification. The mean Ca content of the

aboveground parts of these plants was 19.67 g kg$^{-1}$ (range: 4.34–40.24 g kg$^{-1}$), while the mean Ca content of the underground parts was 10.79 g kg$^{-1}$ (range: 4.41–33.62 g kg$^{-1}$). The Ca content of the aboveground parts was significantly higher than that of the underground parts ($p<0.05$) when compared across the same grades of rocky desertification. Whether the Ca content of aboveground part of the plants or that of underground parts, there were no significant differences ($p>0.05$) among the three different grades of rocky desertification. Furthermore, the grades of rocky desertification had no obvious effect on the Ca content of the aboveground and underground parts of the plants generally (Fig. 2).

### 3.2.2 Ca content in different plant functional groups

The 41 plant species were identified in the 36 small quadrats; these plants were divided into different functional groups. For each functional group, Ca content between the aboveground and underground parts were significantly different ($p<0.05$), and the Ca content of the aboveground parts was higher than that of the underground parts ($p<0.05$) (Fig. 3).

Nitrogen-fixing plants (22.48 g kg$^{-1}$) showed a slightly higher Ca content in the aboveground parts than non-nitrogen-fixing plants (19.39 g kg$^{-1}$; $p>0.05$), although Ca content in the underground parts of nitrogen-fixing plants (6.76 g kg$^{-1}$) was lower than that of non-nitrogen-fixing plants (11.12 g kg$^{-1}$; $p>0.05$). For C3 plants, Ca content in the aboveground and underground parts were 21.08 g kg$^{-1}$, and 13.18 g kg$^{-1}$, respectively, and were both significantly higher than that of C4 plants (aboveground: 15.68 g kg$^{-1}$; underground: 6.42 g kg$^{-1}$; $p<0.05$). In terms of life form functional groups, shrubs showed a significantly higher Ca content, both aboveground and underground than herbs ($p<0.05$), although there were no significant differences ($p>0.05$) between deciduous and evergreen shrubs ($p>0.05$). There was no statistical difference with this respect between annual herbs and perennial herbs ($p>0.05$). The aboveground and underground Ca content of dicotyledons were 21.39 g kg$^{-1}$ and 12.19 g kg$^{-1}$, respectively, and were significantly higher than that of monocotyledons (9.63 g kg$^{-1}$ and 4.79 g kg$^{-1}$, respectively; $p<0.05$).

In terms of monocotyledons and dicotyledons, further analysis revealed no significant differences in the Ca content of the aboveground parts when compared between the different grades of rocky desertification; this was also true for the Ca content of the underground parts. The Ca content of both the aboveground and underground parts of monocotyledons was always low

while those of dicotyledons were always high. The Ca content of dicotyledons was significantly higher than those of monocotyledons across the three grades of rocky desertification ($p<0.05$) (Fig. 4).

For the 41 common plants collected, 17 plant species (which exist in each sample area) were widespread throughout the southwestern rocky desertification areas of Hunan. For each of these 17 species, we calculated their important values (*IV*) (Table. 2). These plants were common species in the local area. We carried out two-way ANOVA for both species and soil for these 17 plants to determine differences in plant Ca content. The soil was graded into three categories: LRD, MRD and IRD. Data showed that the Ca content in the aboveground parts of the 17 plant species were highly significant (df=16, F=11.277, $p<0.01$) among species, although these differences were not significant among the different grades of rocky desertification (df=2, F=2.299, p=0.117). For Ca content in the underground parts, differences were highly significant not only in terms of plant species (df=16, F=8.543, $p<0.01$), but also among the different grades of rocky desertification (df=2, F=4.104, $p<0.01$).

### 3.3 Correlation between plant Ca content and soil ECa content

The correlation between plant Ca content and soil ECa content reflects what extent soil Ca content influences plant Ca content, and may also reflect how different plants respond to differences in soil ECa content. For these 17 plant species, the Ca content in the aboveground and underground parts of *Sanguisorba officinalis* had a significant positive correlation ($p<0.01$) with soil ECa content, which indicated that *Sanguisorba officinalis* was affected greatly by soil ECa content. The Ca content in the underground parts of *Dendranthema indicum* ($p<0.05$), and *Castanea henryi*, showed a significant positive correlation ($p<0.01$) with soil ECa content, indicating that the underground parts of these species were also greatly affected by soil ECa content. The Ca content in the aboveground parts of *Themeda japonica* showed a significant positive correlation ($p<0.01$) with soil ECa content, which indicated that the aboveground parts of *Themeda japonica* was also greatly affected by soil ECa content. For the other plants, the Ca content of the aboveground and underground parts did not show a significant positive correlation ($p>0.05$) with soil ECa content (Table. 3).



## 3.4 Capacity of plants adapting to soil high Ca environments

The above 17 kinds of plants were dominant and common species in rocky desertification areas, and were also the representative species that are able to adapt to a high Ca soil environment. These species appear to have a strong capacity to adapt to high Ca environments in rocky desertification areas. The aboveground parts of plants play an important role in physiological metabolism, and their elemental content reflects the physiological and ecological characteristics of plants (Grubb and Edwards, 1982). The capacity of these plants which are able to adapt to high Ca soil environments can be reflected by two indicators: (i) the correlation between Ca content in the aboveground parts of the plants and soil ECa content; (ii) the species differences in terms of the Ca content of the aboveground parts of plants. Thus, based on the above two indicators, we classified these plants into the following groups: Ca-indifferent plants, high-Ca plants and low-Ca plants (Ji et al., 2009).

The Ca-indifferent plants included *Sanguisorba officinalis*, *Castanea henryi*, *Dendranthema indicum* and *Themeda japonica.* For these plants, there was a significant positive correlation between Ca content in the aboveground or underground parts and the soil ECa content. The Ca content of these plants increased or decreased correspondingly with increases or reductions in soil ECa content, but plant growth was not affected by such changes. These plants did not strictly control the absorption and transport of Ca and may be insensitive to the changes of their own Ca content, and their growth was less affected by soil Ca content. In addition, for other plants, the relationship between Ca content in the aboveground and underground parts and soil ECa content did not show a positive correlation, then these plants were divided into high-Ca plants and low-Ca plants, based on the differences in Ca content in the aboveground parts of these plants. High-Ca plants included *Pyracantha fortuneana*, *Rhus chinensis*, *Loropetalum chinense*, *Serissa japonica*, *Glochidion puberum*, *Indigofera tinctoria* and *Aster baccharoides*. The aboveground parts of these plants could maintain a high Ca content (more than 19 g kg$^{-1}$) under conditions of varying ECa content in the soil. Moreover, the physiological activities of these plants had a higher demand for Ca and may have a strong ability to enrich soil Ca. Low-Ca plants included *Abelia chinensis*, *Vitex negundo*, *Smilax china*, *Miscanthus sinensis*, *Artemisia carvifolia* and *Digitaria sanguinalis.* The aboveground parts of these plants could maintain a low Ca content (less than 19 g kg$^{-1}$) under conditions of varying ECa content in the soil. In addition, the physiological activities of these plants had a lower demand for Ca and could alleviate high Ca stress by inhibiting the absorption of Ca

through the root system and its upward transport (Table. 4).

Finally, the different plant functional groups revealed the differences in Ca content (Fig. 2). In some cases, even within the same plant, there was an inconsistent correlation between Ca content in the aboveground and underground parts and the soil ECa content. Collectively, these findings showed that not all plants adapted to soil high Ca environments in the same way, and that they exhibited a variety of adaptive mechanisms.

## 4 Discussion

### 4.1 Dynamics of Ca content in plants and soil

With the grades of rocky desertification increased, the Ca content of soil increases. This indicated that soil Ca content was affected by the grade of rocky desertification. The aboveground parts of plants had a higher Ca content than the underground parts, although there was no significant difference in plant Ca content when compared between aboveground or underground parts ($p>0.05$) across the different grades of rocky desertification. This indicated that the grade of rocky desertification had no obvious effect on the Ca content of the aboveground and underground parts of the plants studied herein. The mean soil ECa content was 1.46 g $kg^{-1}$ in three rocky desertification areas, which was lower than the average ECa content in tobacco-planting soil in Hunan (3.548 g $kg^{-1}$) (Xu et al., 2007). The average ECa content in IRD areas was 3.09 g $kg^{-1}$, which was several times higher than the previously-reported ECa for non-limestone regions in China (Xu et al., 2007). The range of soil ECa content in the study areas is from 0.02(LRD) to 3.92 g $kg^{-1}$ (IRD), with the maximum and minimum being lower than that of soil on Barro Colorado Island, Panama by Messmer et al. (2014). Tanikawa et al. (2017) revealed that concentrations of TCa and ECa were also low at the deeper horizons in the low-acid buffering capacity (ABC) soils, and pointed to differences in both organic layer thickness and soil chemistry as a reason for affecting Ca accumulation of low- and high-ABC stands. Our research shown soil mean TCa and ECa content were the lowest in LRD areas,and the difference of soil TCa and ECa may be caused by bedrock expose rate (the main chemical composition: $CaCO_3$) (Ji et al., 2009). The average calcium content of aboveground parts of plants was 19.67 g • kg-1, which was lower than that of Hunan flue-cured



tobacco (21.93 g $kg^{-1}$) ( Xu et al., 2007). The maximum and minimum calcium content of plant aboveground parts were 41.79

g $kg^{-1}$ and 2.15 g $kg^{-1}$ respectively, and the maximum and minimum calcium content of plant underground parts were 40.14

g $kg^{-1}$ and 0.42 g $kg^{-1}$ respectively, Which is lower than the calcium content of calcareous plants leaves (maximum 85 .13

g $kg^{-1}$, minimum 6.26 g $kg^{-1}$) by Luo et al. (2014). To most plant calcium content, the aboveground part is larger than the

5 underground part, and for a few plants calcium content, the aboveground part is lower than the underground part (such as

*Sanguisorba officinalis*, *Pyracantha fortuneana* and *Castanea henryi*), which is consistent with the findings of Wang et al.

(2014).

**4.2 Correlation between plant Ca content and soil ECa content**

The $Ca^{2+}$ content in plant cells was proportional to soil $Ca^{2+}$. Calcium-rich soils caused cells to absorb more calcium than the

10 cells themselves require   (White and Broadley, 2003). Zou et al. (2010) showed that soil ECa content and leaf calcium

content was extremely significant positive correlation in pot experiment, and our results showed several plants(*Sanguisorba*

*officinalis*, *Dendranthema indicum*, *Castanea henryi* and *Themeda japonica* ) and soil Eca content was a positive correlation,

but most plant calcium content and soil ECa content was not relevant. Two-way ANOVA of species and soil showed that the

Ca content of the aboveground parts of 17 plant species was mainly affected by species-related factors, while the Ca content of

15 the underground parts was affected by both species-related factors and the grade of rocky desertification, which was in

accordance with data reported previously by Ji et al. (2009). The Ca content in the aboveground parts of nitrogen-fixing plants

was significantly higher than that of the underground parts. Since the transport of Ca was mainly one-way (upward), this result

indicated that nitrogen-fixing plants were the most efficient in terms of the upward transport of Ca, and that Ca was mainly

concentrated in the aboveground parts of the plant; these findings were not consistent with those of Ji et al. (2009). In their

paper, Ji et al. (2009) revealed that dicotyledons were the most efficient at the upward transport of Ca, and studied only three

types of plants (pteridophytes, dicotyledons, monocotyledons) that did not include nitrogen-fixing plants, which may be the

reason for the inconsistency of this previous data with our current findings. In terms of the Ca content of monocotyledons, we

found significant differences ($p<0.01$) between the aboveground and underground parts, but the study by Ji et al. (2009)

revealed that these differences were not significant. This may be because most of the monocotyledons collected were low-Ca

plants. Owing to the fact that the aboveground parts of low-Ca plants maintain a lower Ca content for different grades of rocky

desertification, a significant difference was found between the aboveground and underground parts in monocotyledons. In

addition, the Ca content of monocotyledons was lower than that reported for monocotyledons (Ji et al., 2009), indicating that

different individual monocotyledons showed differing abilities to absorb soil Ca.

### 4.3 High calcium adaptation of plants

Over the past decade, progress has been made in identifying the cellular compartments (e.g., endoplasmic reticulum,

chloroplasts and mitochondria) that regulate Ca balance and signal transduction in plants (Müller et al., 2015). In high Ca

environments, the photosynthetic and growth rate of plants may be affected, and a high Ca concentration within the cytoplasm

may lead to death of the plant (Xiang et al., 2003; Li et al., 2006; Ji et al., 2009; Feng et al., 2010). The capacity of plants to

adapt to high Ca environments is mainly reflected in two ways: adaptations of physiological structures and adaptations of

physiological processes (Luo et al., 2012). In terms of adaptations to physiological structures, the most direct way in which

plants can adapt to high Ca environments is to inhibit the plant roots from absorbing Ca and transporting it to the plant's

aboveground parts (Luo et al., 2012). Some plants fix excess Ca by forming calcified deposits in root tissue in order to limit the

upward transport of Ca (Musetti and Favali, 2003). In addition, calcium oxalate crystals in the plant's crystal cells play a role in

regulating plant Ca content (Ilarslan et al., 2001; Pennisi and McConnell, 2001; Volk et al., 2002). In a high Ca environment,

some plants will form calcium oxalate crystal cells in order to fix excess Ca (Moore et al., 2002; Feng et al., 2010).

Furthermore, an active calcium efflux system plays an important role in the adaptation of plants to high Ca environments (Bose

et al., 2011): Excess Ca in plants is exported from mature leaves to the outside, thereby maintaining a lower concentration of

leaf Ca (Musetti and Favali, 2003). The adaptations relating to physiological processes mainly involve two aspects: the

regulation of internal Ca storage and the control of Ca absorption and transport (Luo et al., 2012). The regulation of internal Ca

storage predominantly depends on plasma membrane Ca transport and intracellular Ca storage; collectively these processes

can regulate the intracellular Ca concentration to a lower level (Bowler and Fluhr, 2000). Plants that adapt to high Ca

environments promote excess $Ca^{2+}$ flow through the cytoplasm or store $Ca^{2+}$ in vacuoles via the cytoplasmic $Ca^{2+}$ outflow and

influx system (Shang et al., 2003; Hetherington and Brownlee, 2004; Wang et al., 2006), in order to regulate the concentration

of cytoplasmic $Ca^{2+}$ to a normal level. The normal growth of plants is maintained by the photosynthetic process and by

respiration of the aboveground parts. Therefore, the regulation and control of the concentration of $Ca^{2+}$ in the plant's

aboveground parts is also key in adapting to a high Ca environment (Yin, 2006).

When the Ca content of soil is high, different plants adopt different adaptation strategies. By considering the survival

differences of plant species in high Ca environments such as lime soil, Simpson (1938) and Tu (1995) divided plants into

non-calcium plants and calcicole. The former is only distributed in acidic soils and other areas with low concentrations of $Ca^{2+}$;

these plants are also known as calcifugous plants or calcifuges (Simpson, 1938; Tu, 1995). According to the level of calcicole

dependence on high Ca environments, these plants can be further divided into non-specific calcicole and specific calcicole. Of

these, specific calcicole are only found within a carbonate matrix and calcareous soil; furthermore, these plants are specific to

the soil environment (Tu, 1995). Depending on their Ca content, non-specific calcicoles can be divided into calciphiles,

calciphilous plants, Ca-indifferent plants (Tu, 1995; Zhou, 1997). The adaptability of plants to high Ca soil environments is

related to their ability to absorb, transport and accumulate Ca (Ji et al., 2009).

The aboveground parts of a plant represent the main site of its physiological activity. Thus, the Ca content in the

aboveground part reflects the Ca demand of the plant's physiological activity. The research conducted by Ji et al. (2009) was

based on the differences in correlation between the Ca content of the aboveground parts of plants and its soil Ca content; these

authors analyzed the capacity of plants to adapt to high Ca environments, and divided the dominated species into Ca-indifferent

plants, high-Ca plants and low-Ca plants. In the present paper, we used this classification method to categorize our plants

species, which were widely distributed across our study environments, thus providing theoretical guidance for vegetation

restoration in rocky desertification areas.

**5 Conclusion**

Our results indicated that the mean soil TCa and ECa content were highest in areas of IRD, followed by MRD, followed by

LRD. Significant differences were detected for both soil ECa and TCa content when compared between the rocky side and non-rocky side of each grade of rocky desertification areas. The Ca content in the aboveground parts of plants was significantly higher than that in the underground parts for the three grades of rocky desertification studied. Furthermore, Significant differences in Ca content were found between the aboveground and underground parts of each plant functional group($p<0.05$). The soil ECa content had a significant effect on the Ca content of the underground parts of plants, but had no significant effect on the Ca content of the aboveground parts. Ca-indifferent plants included *Sanguisorba officinalis*, *Castanea henryi*, *Dendranthema indicum* and *Themeda japonica*. For these plants, a significant positive correlation existed between the Ca content in the aboveground or underground parts and the soil ECa content. High-Ca plants included *Pyracantha fortuneana*, *Rhus chinensis*, *Loropetalum chinense*, *Serissa japonica*, *Glochidion puberum*, *Indigofera tinctoria* and *Aster baccharoides*. In this case, the aboveground parts of these plant were able to maintain a higher Ca content under conditions of variable soil ECa content. Finally, low-Ca plants included *Abelia chinensis*, *Vitex negundo*, *Smilax china*, *Miscanthus sinensis*, *Artemisia carvifolia* and *Digitaria sanguinalis*. The aboveground parts of low-Ca plants were able to maintain a lower Ca content under conditions of variable soil ECa content.

**Acknowledgements**

This work was supported by the Forestry Science and Technology Promotion Project of State Forestry Administration of China ([2014]52), and the Desertification (Rocky Desertification) Monitoring Project of State Forestry Administration of China (20150618 and 20160603).



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





**Table 1. Soil TCa and ECa content from different grades of rocky desertification.**

| Ca typical (g kg$^{-1}$) | Sample point | LRD | MRD | IRD |
|---|---|---|---|---|
| TCa | Non-rocky side | 1.19±0.45Aa | 2.33±0.53Ba | 2.62±0.97Ba |
| | Rocky side | 1.68±0.53Ab | 2.97±0.29Bb | 5.66±1.37Cb |
| | Average | 1.31±0.51A | 2.53±0.56B | 3.38±1.71B |
| ECa | Non-rocky side | 0.51±0.26Aa | 1.68±0.37Ba | 1.63±0.88Ba |
| | Rocky side | 0.97±0.39Ab | 2.20±0.39Bb | 3.09±0.58Cb |
| | Average | 0.63±0.36A | 1.83±0.44B | 2.00±1.03C |
| Ca effectiveness | ECa/TCa (%) | 47.72 | 72.55 | 58.98 |

Data represent mean ± standard deviation. Different lower-case letters in each column represent significant differences in different sample points within the same grade of rocky desertification. Different upper-case letters in each row represent significant differences between different grades of rocky desertification ($p < 0.05$).





**Table 2. The main species of plant identified during this study and their important value (*IV*) in different grades of rocky desertification.**

| Vegetable layer | Species | Important Value (*IV*) | | |
|---|---|---|---|---|
| | | LRD(%) | MRD(%) | IRD(%) |
| Shrubs | *Abelia chinensis* | 18.56 | 6.91 | 21.65 |
| | *Castanea henryi* | 22.33 | 1.35 | 5.32 |
| | *Indigofera tinctoria* | 5.10 | 16.64 | 4.30 |
| | *Pyracantha fortuneana* | 5.26 | 4.83 | 1.63 |
| | *Loropetalum chinense* | - | 1.00 | 10.45 |
| | *Serissa japonica* | 4.13 | 5.80 | 7.45 |
| | *Vitex negundo* | 4.85 | 11.38 | 19.07 |
| | *Rhus chinensis* | 0.84 | 7.11 | 2.24 |
| | *Smilax china* | - | 1.23 | 1.02 |
| | *Glochidion puberum* | 11.36 | 4.81 | 4.19 |
| | *Ilex chinensis* | 2.25 | - | - |
| | *Ilex cornuta* | - | - | 1.32 |
| | *Elaeagnus pungens* | - | 1.70 | - |
| | *Lespedeza bicolor* | 3.01 | 0.58 | - |
| | *Symplocos chinensis* | 2.07 | - | 1.57 |
| | *Broussonetia kaempferi* | - | 0.79 | - |
| | *Populus adenopoda* | 1.06 | - | - |
| Herbs | *Miscanthus sinensis* | 36.54 | 5.82 | 36.36 |
| | *Artemisia carvifolia* | 17.38 | 9.04 | 14.02 |
| | *Sanguisorba officinalis* | 1.41 | 1.01 | 2.14 |
| | *Themeda japonica* | 1.85 | 18.23 | 5.03 |
| | *Dendranthema indicum* | 3.82 | 16.94 | 6.55 |
| | *Digitaria sanguinalis* | 6.83 | 3.95 | 10.57 |
| | *Aster baccharoides* | 2.40 | - | 4.30 |
| | *Imperata cylindrica* | - | 3.30 | - |
| | *Salvia plebeia* | - | - | 0.81 |
| | *Patrinia scabiosaefolia* | 0.29 | - | - |
| | *Sonchus arvensis* | - | - | 0.51 |

"-" indicates that the important value (*IV*) of these species are less than 1.



**Table 3. Correlations between the Ca content of 17 plant species and the soil ECa content of different rocky desertification areas.**

| Species | Ca content in aboveground parts | | | Ca content in underground parts | | |
|---|---|---|---|---|---|---|
| | Range (g kg$^{-1}$) | Mean±SE (g kg$^{-1}$) | Correlation coefficient | Range (g kg$^{-1}$) | Mean±SE (g kg$^{-1}$) | Correlation coefficient |
| *Smilax china* | 5.77~36.35 | 18.5±12.24 | 0.302 | 3.11~8.61 | 5.89±2.75 | 0.931 |
| *Aster baccharoides* | 16.16~24.03 | 20±3.6 | 0.418 | 6.2~12.02 | 8.91±2.58 | 0.315 |
| *Vitex negundo* | 5.53~26.31 | 18.03±7.44 | 0.198 | 2.83~8.17 | 5.59±2.02 | −0.116 |
| *Sanguisorba officinalis* | 17.68~27.77 | 24.01±4.47 | 0.995** | 13.41~40.14 | 32.25±12.71 | 0.996** |
| *Themeda japonica* | 2.15~9.23 | 5.51±2.45 | 0.963** | 0.42~7.91 | 3.88±2.7 | 0.488 |
| *Pyracantha fortuneana* | 9.16~29.84 | 19.61±8.46 | 0.240 | 17.08~31.86 | 21.43±7.02 | −0.189 |
| *Loropetalum chinense* | 10.33~33.44 | 27.25±7.29 | −0.203 | 13.62~27.69 | 19.69±7.09 | 0.542 |
| *Serissa japonica* | 9.69~33.66 | 23.26±9.9 | −0.027 | 4.27~20.51 | 12.01±7.81 | 0.838 |
| *Indigofera tinctoria* | 10.18~40.24 | 24.17±11.49 | 0.215 | 3.39~9.83 | 5.98±2.33 | −0.289 |
| *Digitaria sanguinalis* | 4.75~9.8 | 6.67±2.73 | 0.257 | 1.36~5.33 | 3.37±1.98 | −0.915 |
| *Abelia chinensis* | 5.07~29.64 | 18.08±10.12 | −0.163 | 0.87~7.12 | 4.1±2.16 | 0.070 |
| *Artemisia carvifolia* | 15.34~19.39 | 17.37±1.42 | 0.400 | 6.39~14.07 | 9.18±3.07 | 0.028 |
| *Glochidion puberum* | 11.13~26.99 | 20.49±7.04 | 0.357 | 5.33~13.64 | 10.45±4.48 | 0.775 |
| *Miscanthus sinensis* | 4.34~7.6 | 5.61±1.44 | 0.000 | 2.88~13.1 | 5.82±4.87 | 0.118 |
| *Rhus chinensis* | 10.52~28.16 | 19.93±6.43 | 0.076 | 8.92~20.38 | 14.13±4.13 | 0.336 |
| *Dendranthema indicum* | 20.97~24.96 | 22.54±1.86 | 0.666 | 2.97~7.39 | 5.39±1.7 | 0.877* |
| *Castanea henryi* | 12.99~38.74 | 22.4±8.17 | 0.151 | 20.52~31.37 | 25.28±3.92 | 0.963** |

Coefficients are significant at *p* < 0.05 (*) and < 0.01 (**).





**Table 4. Adaptation of plants to high Ca environments in rocky desertification areas.**

| Types of adaptation | Species | Characteristics of calcium content in plants | Strategies of plant adaptation to high calcium environments |
|---|---|---|---|
| Ca-indifferent plants | *Sanguisorba officinalis* *Castanea henryi* *Dendranthema indicum* *Themeda japonica* | There is significant positive correlation between the calcium content in the aboveground/underground parts of plants and the soil ECa content. The coefficient of variation for calcium content in plants has a wide range. | Plants adapt to different calcium contents in soil through high $Ca^{2+}$ buffering capacity. By regulating $Ca^{2+}$ binding in calcium stores, the $Ca^{2+}$ concentration in cytoplasm is maintained at a stable level. |
| High-Ca plants | *Loropetalum chinense* *Serissa japonica* *Indigofera tinctoria* *Glochidion puberum* *Aster baccharoides* *Pyracantha fortuneana* *Rhus chinensis* | There is no significant positive correlation between the calcium content in the aboveground parts of plants and the soil ECa content. The aboveground part has a high level of calcium content and the coefficient of variation falls within a narrow range. | Plants maintain high calcium content by enhancing calcium uptake and transporting it from underground to aboveground parts. High calcium is needed or tolerated in these plants. |
| Low-Ca plants | *Vitex negundo* *Abelia chinensis* *Smilax china* *Miscanthus sinensis* *Artemisia carvifolia* *Digitaria sanguinalis* | There is no significant positive correlation between the calcium content in the aboveground parts of plants and the soil ECa content. The aboveground part has a low level of calcium content and the coefficient of variation falls within a narrow range. | Plants maintain low calcium content in the aboveground parts by reducing calcium uptake and transporting it from underground to aboveground parts. |



## Figure captions

**Fig. 1 Geographical locations of the study sites.**

**Fig. 2 Characteristics of plants Ca content in different grades of rocky desertification.** LRD, light rocky desertification; MRD, moderate rocky desertification; IRD, intense rocky desertification. Different lower-case letters represent significant differences in the Ca content between the aboveground and underground parts of the plants in the same grade of rocky desertification; different upper-case letters represent significant differences in the Ca content of the plants among the different grades of rocky desertification ($p<0.05$).

**Fig. 3 Ca content in the aboveground and underground parts of plants in different functional groups of plans.** Different lower-case letters represent significant differences between the Ca content of the aboveground and underground parts for the same functional groups ($p<0.05$); different upper-case letters represent significant differences among different functional groups, ($p<0.05$). Annual herbs include annual and biennial herbs, while deciduous shrubs include deciduous trees with a height < 2 m or a ground diameter < 3 cm.

**Fig. 4 Ca content in the aboveground and underground parts of different plant types from three different rocky desertification sample areas.** LRD, light rocky desertification; MRD, moderate rocky desertification; IRD, intense rocky desertification. Values with the same letters were not significantly different ($p>0.05$).



**Fig. 1**





**Fig. 2**

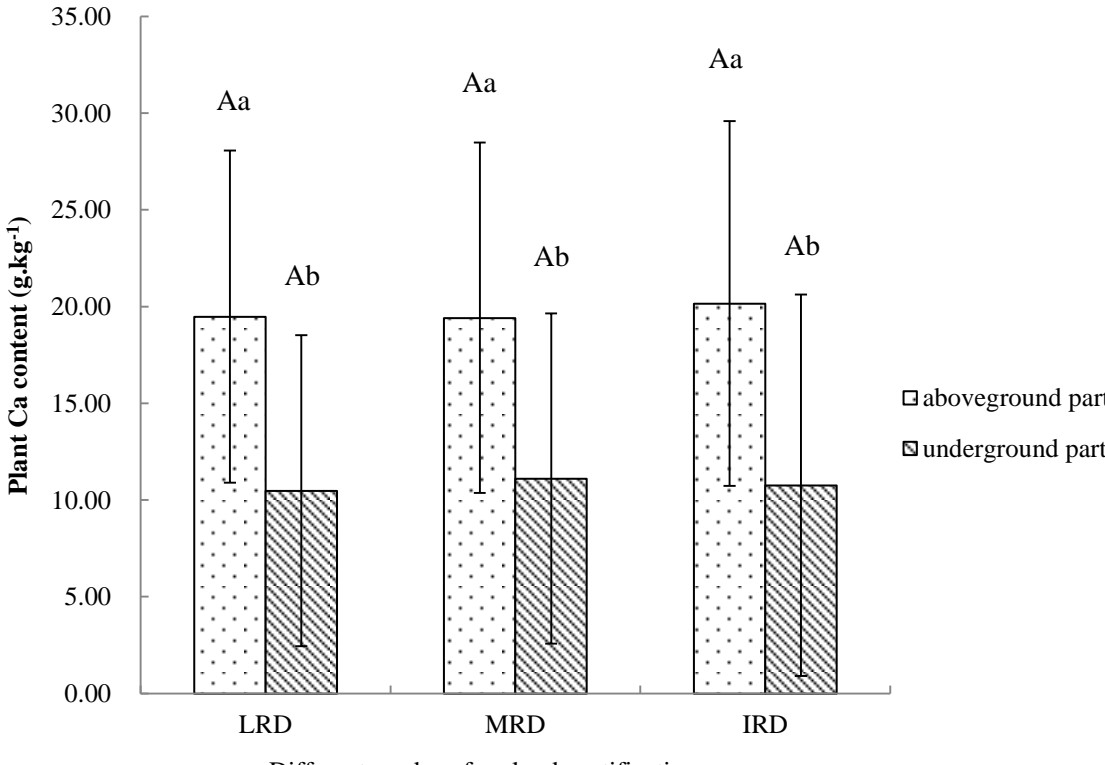





**Fig. 3**

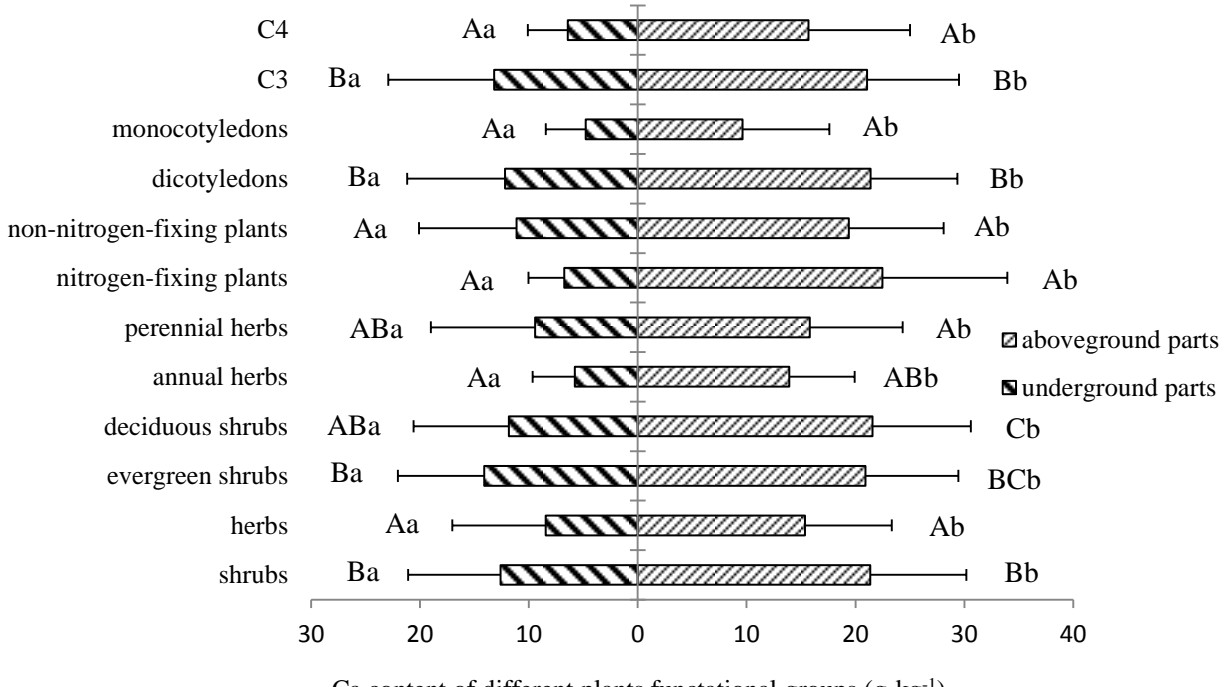

Ca content of different plants functational groups (g kg$^{-1}$)



**Fig. 4**

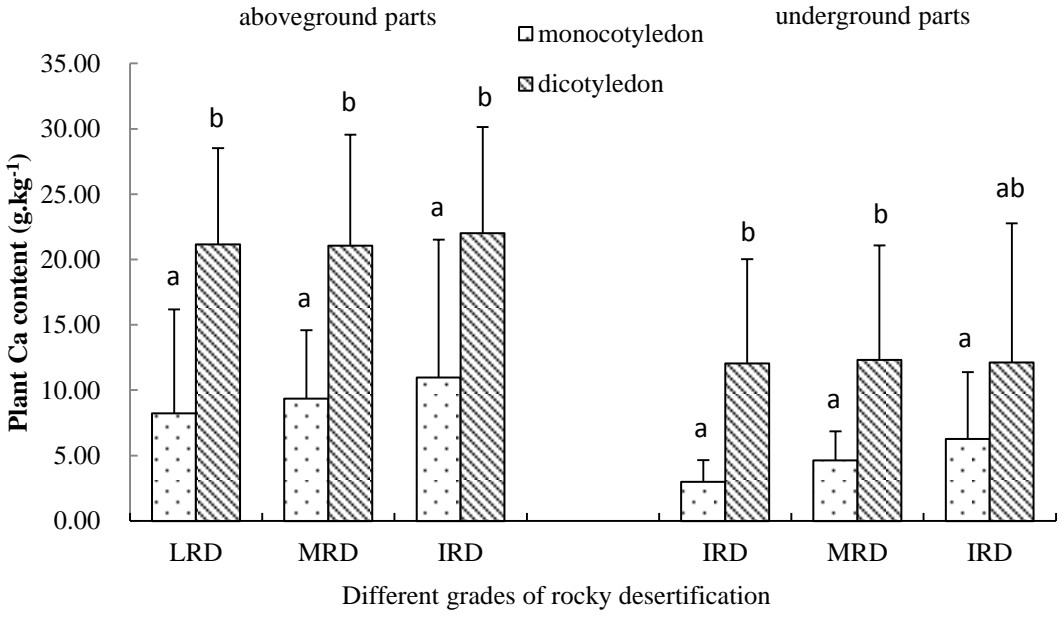



**Author contributions**

Idea and study design: Deng X. W., Wei X. C.; Experiments and statistical analysis: Deng X. W., Wei X. C., and Wen H. F.;

Manuscript writing: Wei X. C.; Discussion and revision: Xiang W. H., Ouyang S., Lei P. F., Chen L. All authors have read

and approved the content of the manuscript.

5    **Competing interests**

The authors declare that they have no conflict of interest.