# Peer review of "Calcium content and high calcium adaptation of plants in karst areas of southwestern Hunan, China"

_Biogeosciences, 2017_

## Referee Comment (RC1) · Anonymous Referee #1 · 3 Jan 2018

My comment

The manuscript, "Calcium content and high calcium adaptation of plants in karst areas of southwestern Hunan, China" was intended to investigate plant Ca content, soil exchangeable Ca and total Ca content on the rocky desertification areas in southwestern China. The present project is important and interested for us to better understanding of differences of Ca content in plants and soil resulted from the grades of rocky desertification. The contents of this manuscript is meet the scopes of the Journal of "Biogeosciences" for publication very well. However, there are some shortcomings in the manuscript, which prevents it from publishing the present version in this journal.

[Figure]

The structure and organization of sentences in the manuscript are not appropriately written, which are confused by readers. The Results and Discussion sections should not be mixed together in writing and these two sections should be separated in the manuscript. Authors should not present any explanations in the Results section, and the similar errors are appeared in the Discussion section. Therefore, I suggest the manuscript be accepted after major revise. I provide my comments in detail as following for consideration when the manuscript is revised.

1. Abstract: P1 L13 removes the sentence "However, the Ca2+ dynamics of plants and soil are not well understood" into the P4 L10 P1 L14 deletes the s from the "samples" P1 L16 "slop" should be slope Reword all words "underground" into belowground in the entire manuscript

2. Introduction P2 L2: use provides to replace "can provide" P2 L8: delete "Of course" Change the sentence "the severity of rocky desertification in Hunan Province was ranked fourth (Li et al., 2016) intoïijŻThe severity of rocky desertification was ranked in fourth in Hunan Province of China (Li et al., 2016) P2 L9: Insert the "Rocky desertification is an extreme form of land degradation in karst areas, and 10 has become a major social problem in terms of China's economic and social development (Sheng et al., 2015)" should be before "The severity of rocky desertification was ranked in fourth in Hunan Province of China (Li et al., 2016)" P2 L110-12: Change the "The restoration and reconstruction of rocky desertification ecosystems has become the immediate focus of agro-forestry production environment improvements, regional economic development and helping to support people out of poverty (Jing et al., 2016)" into "The restoration and reconstruction of rocky desertification ecosystems have become the urgent environment improvements, regional economic development by using agroforestry system and helping to support people out of poverty (Jing et al., 2016). P2L13 "soil with high Ca" P2L14-15: rewrite "From the origin of rocky desertification, 15 the restoration of vegetation is key to the process of remediation (Wang et al., 2004). Consequently, the screening of plants which can grow successfully in high- calcium environments is

an extremely critical step." P3L2: Change "Ca2+ and pectin in the cell walls of plants combine" into: "Ca2+ combine with pectin in the cell walls of plants" P3L11: change" than cannot" into "not" P4L3-L4 "Ji et al. (2009) revealed that the mean soil ECa was 3.61 gÂůkg−1 in the Puding, Huajing, Libo and Luodian Counties of Guizhou Province, which is several times that of non-limestone areas in China." should be: "The mean soil ECa was 3.61 gÂůkg−1 in the Puding, Huajing, Libo and Luodian Counties of Guizhou Province, which is several times that of non-limestone areas in China (Ji et al. 2009). P4L7-L19 "These results indicate that there are differences in soil Ca content between different areas and that there are differences between calcareous and non-calcareous plants in terms of Ca absorption, transport and storage and other physiological processes. Collectively, these differences lead to different degrees of adaptability of plants to high Ca environments." Should be "There are variations in soil Ca content among different areas and differences between calcareous and non-calcareous plants in terms of Ca absorption, transport and storage and other physiological processes. These differences need to identify the variety of the plants to adapt with high Ca environments." P4L10-L11 delete "there is a scarcity of extensive research into" should be" the mechanisms by which plants adapt to high Ca conditions, particularly in karst areas and the Ca2+ dynamics of plants and soil are not well understood. P4L14: delete "In order to", capitalize the "To" P4L15 "we did the following:" should be "the following investigations were explored"

3. Materials and methods P5: site description is too simple, should add more information regarding to the study, e.g. slope, soil pH, soil properties, and vegetation cover P5L4 title "Data collection" should be "Experimental design and data collection" P5L6 delete "period. These four main indices" P5L9-L10: delete" a range of 10 characteristics and data relating to the surrounding environment", add "environmental factors" P5L11: "We conducted a detailed survey of the three sample areas and collected samples in October 2016." Should be "The sample collection in these three sample areas were conducted in October 2016. " P5L13: use "Within" to replace the "For" P5L14: add were set up after "(upper, middle, and lower slope)" P5L14-L15: add (3x4x3) were set

up before the "for analysis", delete "We chose to study" and "the common plant species of the region, 15 and gathered plants using the whole plant harvest method. In each small quadrat, every kind of shrubs and herbs are collected." Should be "The common plant species of the region were gathered using the whole plant harvest method in each small quadrat as well as all shrubs and herbs are collected." P5L17: "heated" should be "oven tried' P5L18: add "and after" after the de-enzyme" P5L18: not clear, "constant weight at 80oC, L 17 you mentioned 105oC, why? Rewrite it P5L18: delete "and bagged" and add "chemical" before the "analysis" P5L20: delete "Finally" add "were sampled" P6L5: "biennial herbs, while 'deciduous shrubs' included deciduous trees with a height less than 2 m or a ground diameter less than 3 cm." not clear, rewrite P6L8-L9: "One-way analysis of variance (ANOVA), Two-way ANOVA and Pearson correlation analysis ($\alpha$ = 0.05) were used to analyze the Ca content of soil and plants within and between different grades of rocky desertification." Not clear, rewrite it

4. Results P6L13ïijŽadd in soil after "The mean TCa content" P6L14: Use "location" to replace "points" P6L15: delete "Furthermore", and add "The", and to use "that" replacing "to be" P6L17: Add "Ca content" after: "average", and use "the" to replace "with" P6L21: Use "Total" to replace "The" P7L1: use "." To replace ",", and then use "Compare to" to replace "while" P7L3: delete "when compared across", to use" throughout" P7L4: "aboveground" add "and belowground", and delete "or that of underground parts, there" P7L5ïijŽDelete the whole sentence "Furthermore, the grades of rocky desertification had no obvious effect on the Ca content of the aboveground and underground parts of the plants generally (Fig. 2)." P7L8: "The 41 plant species were identified in the 36 small quadrats; these plants were divided into different functional groups" should be "The 41 plant species were identified and were divided into different functional groups in the 36 small quadrats." P7L8-L9: delete "For each functional group," add "The" before Ca P7L9-L10: "Ca content between the aboveground and underground parts were significantly different (p< 0.05), and 10 the Ca content of the aboveground parts was higher than that of the underground parts (p<0.05)" should beïijĆThe Ca content of the aboveground parts significantly was higher than that of the

belowground parts in each group (p<0.05)" P7L15-L16: "In terms of life form functional groups, shrubs showed a significantly higher Ca content, both aboveground and underground than herbs (p<0.05)" should be "In life form functional groups, shrubs showed a significantly higher in Ca content than herbs in both aboveground and underground (p<0.05)" P7L18: "The aboveground and underground Ca content of dicotyledons" should be" The Ca content of dicotyledons in aboveground and belowground parts" P7L21-23: "In terms of monocotyledons and dicotyledons, further analysis revealed no significant differences in the Ca content of the aboveground parts when compared between the different grades of rocky desertification; this was also true for the Ca content of the underground parts." Its not clear, rewrite it P7L23: Delete "The Ca content of both the aboveground and underground parts of monocotyledons was always low while those of dicotyledons were always high" P8L1-2 "The Ca content of dicotyledons was significantly higher than those of monocotyledons across" should be "The Ca content of dicotyledons in both of was significantly higher than those of monocotyledons in both aboveground and belowground parts throughout" P8L3: "For the 41 common plants collected, 17 plant species (which exist in each sample area) were widespread throughout the southwestern rocky desertification areas of Hunan." Should be "Within total 41 common plants species, 17 plant species were found in each sample plot and were widespread throughout the southwestern rocky desertification areas of Hunan." P8L5: Delete "For each of t", capitalize "T" P8L3: use were calculated replace "we calculated" P8L5: Delete ". These plants were common species in the local area" P8L5-"We carried out two-way ANOVA for both species and soil for these 17 plants to determine differences in plant Ca content" should be moved to the data analysis part, not in the results part P8L6: Delete ". The soil was graded into three categories: LRD, MRD and IRD." P8L7: Delete "df=16, F=11.277" P8L8: Use "related among" to replace "significant among the different P9L9: (df=2, F=2.299, p=0.117) P8L9: "The" Ca not For "Ca", delete "differences" P8L10: Use "among the species", delete "(df=16, F=8.543, p<0.01)", and delete "but also among the different" and it throughout all the " , and delete "df=2, F=4.104," P8L12-13: "The correlation between plant Ca content and

soil ECa content reflects what extent soil Ca content influences plant Ca content, and may also reflect how different plants respond to differences in soil ECa content" this sentence should not in results part, may be in discussion section. P8l13: Too many "For this" and "In terms of", delete them. P8L15: "which indicated that Sanguisorba officinalis was affected greatly by soil ECa conten" should be not in the results section. P8L17: "indicating that the underground parts of these species were also greatly affected by soil ECa content." should be not in the results section. P8L19" which indicated that the aboveground parts of Themeda japonica was also greatly affected by soil ECa 20 content" should be not in the results section P9L2: Delete "kinds of" P9L3: "and were also the representative species that are able to adapt to a high Ca soil environment." How do you know it? Suggest to delete it P9L6-9: "The capacity of these plants which are able to adapt to high Ca soil environments can be reflected by two indicators: (i) the correlation between Ca content in the aboveground parts of the plants and soil ECa content; (ii) the species differences in terms of the Ca content of the aboveground parts of plants. Thus, based on the above two indicators, we classified these plants into the following groups: Ca-indifferent plants, high-Ca plants and low-Ca plants (Ji et al., 2009)." This should be moved to the discussion section. Results just present your results, no explanation and citation. P9L10: The definition "Ca-indifferent plants" is it correct? P9L12:" The Ca content of these plants increased or decreased correspondingly with increases or reductions in soil ECa content, but plant growth was not affected by such changes." Not clear, rewrite it P9L17: "High-Ca plants", refer it "high Ca demand plants P9L20: "Moreover, the physiological activities of these plants had a higher demand for Ca and may have a strong ability to enrich soil Ca." should be in the discussion part P9L21: "Low-Ca plants" should be Low Ca demand plants P9L23: Why do you use "19g/kg" as the boundary to determine the low or high Ca demand plants? P10L2-5: the whole paragraph should belong to the discussion, not in the results. Again, the results should just present your results, do not need any explanation in this part, any explanation and citation should be in the discussion section.

5. Discussion P10L9: delete "The aboveground parts of plants had a higher Ca con-

tent than the underground 10 parts, although" P10L10: Capitalize T (The), delete when compared P10L17: period after the (2014), and then separate the paragraph P10L18: What is the ABC soil? P10L18: "Tanikawa et al. (2017) revealed that concentrations of TCa and ECa were also low at the deeper horizons in the low-acid buffering capacity ïijĹABCïijĽ soils, and pointed to differences in both organic layer thickness and soil chemistry as a reason for affecting Ca accumulation of low- and 20 high-ABC stands" is unclear, rewrite it P11L1: Add "compared to the aboveground and belowground Ca content in our study," before the "The", and then use lowercase of "t" P11L3: "," should be "." P11L1-4: "The maximum and minimum calcium content of plant aboveground parts were 41.79 gÂůkg−1 and 2.15 gÂůkg−1 respectively, and the maximum and minimum calcium content of plant underground parts were 40.14 gÂůkg−1 and 0.42 gÂůkg−1 respectively, Which is lower than the calcium content of calcareous plants leaves (maximum 85 .13 gÂůkg−1ïijŇminimum 6.26 gÂůkg−1) by Luo et al. (2014)." Aboveground includes leaves and branches, how do you compared with leaves only? Ca presents the Calcium, should keep the constant in the manuscript. P11L9: The beginning of the paragraph should present your research results pattern first, and then discuss and explain it. P11L11: Use "had a" to replace "was extremely". Use "." and delete "and" to separate the sentence. The sentence "our results showed several plants (Sanguisorba officinalis, Dendranthema indicum, Castanea henryi and Themeda japonica ) and soil Eca content was a positive correlation, but most plant calcium content and soil ECa content was not relevant." Should be "Our results showed that most plants had no correlation relationship between soil ECa and plant Ca excepting several plant (Sanguisorba officinalis, Dendranthema indicum, Castanea henryi and Themeda japonica ) had a positive correlation between soil Eca and plant Ca content." P11L14: what are "species-related factors,"? Do you mean plant species physiological factors? P11L15-16: "was in accordance with data reported previously by Ji et al. (2009)." should be "was supported with data reported by Ji et al. (2009)." P11L17: "Since the transport of Ca was mainly one-way (upward), this result indicated that nitrogen-fixing plants were the most efficient in terms of the upward transport of Ca,

and that Ca was mainly concentrated in the aboveground parts of the plant; these find-ings were not consistent with those of Ji et al. (2009)." Is not clear, rewrite it. P11L19: delete "In the paper" P11L20: "." after Ca, The sentence" and studied only three types of plants (pteridophytes, dicotyledons, monocotyledons) that did not include nitrogen-fixing plants, which may be the reason for the inconsistency of this previous data with our current findings." Should be" They used three types of plants (pteridophytes, di-cotyledons, monocotyledons) exclude nitrogen-fixing plants in their study, which may have a conflicting result compared with our current findings." P11L22-P12L1: delete "in terms of", the sentence "In terms of the Ca content of monocotyledons, we found significant differences (p<0.01) between the aboveground and underground parts, but the study by Ji et al. (2009) revealed that these differences were not significant. This may be because most of the monocotyledons collected were low-Ca plants." should be"We found significant differences (p<0.01) between the aboveground and below-ground parts in Ca content of monocotyledons in our study. However, Ji et al. (2009) revealed that no significant differences between the aboveground and belowground parts in Ca content of monocotyledons. This phenomenon may contribute the most of the monocotyledons sample plants were low-Ca demand plants." P12L2-3: "Owing to the fact that the aboveground parts of low-Ca plants maintain a lower Ca content for different grades of rocky desertification, a significant difference was found between the aboveground and underground parts in monocotyledons. In addition, the Ca content of monocotyledons was lower than that reported for monocotyledons (Ji et al., 2009)," is not clear, rewrite it P12L7-8: "Over the past decade, progress has been made in identifying the cellular compartments (e.g., endoplasmic reticulum, chloroplasts and mitochondria) that regulate Ca balance and signal transduction in plants (Müller et al., 2015). " may move to the introduction section. P12L8-P13L14: again, authors should present the results pattern at the beginning of the discussion to explain your results. This paragraph should be rewritten. Lots of "in terms of" showed in the manuscript, delete them. In this paragraph, I did not see any results at the beginning of the dis-cussion. The discussion is used to explain the results P13L15-22: suggest deleting

the paragraph because it does not make sense in your discussion, as well as so many times to cite the literature Ji et al (2009)

6. Conclusions P13L5 "Conclusions" P1323-P14L1: delete "followed by" add "and" before "LRD" P14L1-2: delete the sentence" Significant differences were detected for both soil ECa and TCa content when compared between the rocky side and non-rocky side of each grade of rocky desertification areas. " P14L3: add "sites" after "studied", delete "Furthermore" P14L5: Delete "(p<0.05) L14L6ïijŻDelete "the" Ca P14L6: Ca-indifferent" is correct? P14L7: "," after "Themeda japonica", delete "For these plants," and put had P14L8: "High-Ca plants included Pyracantha fortuneana," should be" High-Ca plants in our study were Pyracantha fortuneana, P14L10: delete "In this case"ïijŇthe sentence "the aboveground parts of these plant were able to maintain a higher Ca content under conditions of variable soil ECa content. "should be" the aboveground parts of these plant were able to absorb a high Ca content from various of ECa content soils.

Please also note the supplement to this comment:
https://www.biogeosciences-discuss.net/bg-2017-392/bg-2017-392-RC1-supplement.pdf

---

## Author Comment (AC2) · 13 Jan 2018

We highly appreciate the valuable comments and suggestions by the anonymous reviewer on our manuscript. We have attempted to address each point raised by the reviewer. The following Supplement file is our edited manuscript according to the comments by the reviewer.

Please also note the supplement to this comment:
https://www.biogeosciences-discuss.net/bg-2017-392/bg-2017-392-AC2-supplement.zip

---

## Referee Comment (RC2) · Anonymous Referee #2 · 18 Jan 2018

My comments

1. General comments

The authors' manuscript "Calcium content and high calcium adaptation of plants in karst areas of southwestern Hunan, China" introduced an investigation and an analysis of the relationships between the degree of rocky desertification and calcium content in soil and plant. The author's results are interesting. These results can be seen as a valuable reference, which could be helpful in related research works to screen plant species for vegetation restoration in karst areas of China. As for the study itself, this paper is worthy to be published in the journal "Biogeosciences". However, this

manuscript needs a major revision before publication, especially revisions in paper structure and writing quality.

2. Specific comments

Because there is no line number in the manuscript that I downloaded, I suggest the authors put numbers for each line in the revised version if the manuscript goes to the next stage. Here I provide my comments for the sections.

(1) For "Introduction"

In the section of Introduction, I cannot find sufficient reviews or introductions of existing studies on the issue addressed by the authors' study. There are only two existing studies (Zhang 2005; Ji et al., 2009) mentioned in this section. The study background need be introduced more in this section. I wonder if the authors can provide a brief review for this issue including some important related studies reported for other countries. Readers may want to know whether the authors' hypothesis, "the dynamics of Ca content is significantly affected by the grade of rocky desertification", is supported by more studies or not.

The second, third, and fourth paragraphs are lengthy in describing the knowledge of plant physiology. I suggest shortening them to several sentences for outlining some key processes.

(2) For "Results"

The major problem is that there are lots of explanations and analyses in this section, especially in section 3.2.2 and 3.4. Of course, for a better understanding, it may be ok to arrange a few explanations in "Results". But any analysis should not appear in the "Results". Otherwise, this section can be "Results and discussions".

(3) For "Discussions"

Overall, there are still too much knowledge descriptions of plant physiology. I think

that only the necessary knowledge should be mentioned corresponding to the new findings, instead of a detailed introduction of knowledge. I would rather see that the discussions focus on the main points and your hypothesis, or analyze your main work, for example, the three parts of your work: (i) to measure the soil Ca contents; (ii) to compare between above- and below-ground parts of plants; and (iii) to analyze correlation between Ca in plants and soils.

For section 4.1:

According to the text, the authors may discuss their works (i) and (ii) in section 4.1. However, section 4.1 seems not well organized. Actually, the results have already shown the dynamics of soil Ca, and the difference between above- and below-ground parts of plants. I hope the section 4.1 can summarize these two works clearly, and can indicate some new findings. In addition, for readability, readers may need to know percentages of your measurements to the values reported by other studies. I also suggest the authors consider changing the title from "Dynamics of Ca content in plants and soil" to "Dynamics and ranges of Ca content in plants and soil" since most parts in this section are talking about the "ranges".

For section 4.2:

The authors' work (iii) was discussed in this section. It seems to me, the content needs to be reorganized very logically. Section 4.2 lists some results and other researchers' conclusions, however, the logical relationships between those results and conclusions are not clear. For example, the first sentence states "The $Ca^{2+}$ content in plant cells was proportional to soil $Ca^{2+}$". Then what parts of plants are you talking about? Above-ground, below-ground, or whole tree?

The second sentence states "Calcium-rich soils caused cells to absorb more calcium than the 10 cells themselves require (White and Broadley, 2003)". Then is this cited sentence for supporting the first sentence, just explaining the cause, or conducting the third sentence? A conjunction word seems necessary.

[Figure]

The third sentence states "Zou et al. (2010) showed that soil ECa content and leaf calcium content [were] extremely significant positive correlation". Is this conclusion cited for comparing your results? If so, it would be better to indicate the plant name(s). Your next statement is "our results showed several plants (......) and soil Eca content was a positive correlation, but most plant calcium content and soil ECa content was not relevant". Zou et al. focused on the leaf, but, do you focus on whole tree? The results and comparisons need to be explained more clearly and logically to avoid reader's confusion.

These above writing issues are raised just for example. The whole section needs to be rewritten for better readability.

For section 4.3:

This section discusses the most important scientific issue (High calcium adaptation of plants) addressed in the study. The solution of this issue may provide useful guidance on vegetation restoration. However, basic knowledge descriptions take up lot of space. I had liked to see the discussions on: (a) Based on study results for the 17 selected species, what are the primary characteristics for each of the three categories (Ca-indifferent plants, high-Ca plants and low-Ca plants)? (b) What should the screening of plant species notice in the vegetation restoration? (c) What are the application prospects in solving the problem of land degradation using the authors' results in karst areas? (d) Is there any unsolved issue, related with this study and remained for further research?

Again, results are interesting and helpful for associated studies. I suggest accepting this manuscript after a major revision. The writing quality should be improved, including a spelling check. As I am not a native English speaker, I will not suggest more regarding language. Good luck!

---

## Author Response (AR1)

**Response to the reviews, list of all relevant changes, and marked-up manuscript**

*Part 1:* **Point-by-point response to Reviewers' Comments**

We highly appreciate the valuable comments and suggestions by the editor and two anonymous reviewers on our manuscript. We have attempted to address each point raised by the editor and the reviewers. The following is our detail responses we have made, with reference to the order of the comments by the editor and the reviewers.

**Reviewer 1:**

**1. Abstract:**

P1 L13 removes the sentence "However, the Ca2+ dynamics of plants and soil are not well understood" into the P4 L15

P1 L14 deletes the s from the "samples"

P1 L16 "slop" should be slope

Reword all words "underground" into belowground in the entire manuscript

Re.1 Thanks for the positive suggestion. And sorry for the mistakes to write "slope" as "slop" and "belowground" is more suitable for our purposes. We have corrected all of them in the entire manuscript.

**2. Introduction**

P2 L2: use provides to replace "can provide"

P2 L8: delete "Of course"

Change the sentence "the severity of rocky desertification in Hunan Province was ranked fourth (Li et al., 2016) into; The severity of rocky desertification was ranked in fourth in Hunan Province of China (Li et al., 2016)

P2 L9: Insert the "Rocky desertification is an extreme form of land degradation in karst areas, and 10 has become a major social problem in terms of China's economic and social development (Sheng et al., 2015)" should be before "The severity of rocky desertification was ranked in fourth in Hunan Province of China (Li et al., 2016)"

Re.2 Thanks, we agree to these advices and the manuscript was revised, accordingly.

P2 L110-12: Change the "The restoration and reconstruction of rocky desertification ecosystems has become the immediate

focus of agro-forestry production environment improvements, regional economic development and helping to support people out of poverty (Jing et al., 2016)" into "The restoration and reconstruction of rocky desertification ecosystems have become the urgent environment improvements, regional economic development by using agroforestry system and helping to support people out of poverty (Jing et al., 2016).

5    P2L13 "soil with high Ca"

**Re.3 Thanks, it is a good suggestion.**

P2L14-15: rewrite "From the origin of rocky desertification, 15 the restoration of vegetation is key to the process of remediation (Wang et al., 2004). Consequently, the screening of plants which can grow successfully in high- calcium

10    environments is an extremely critical step."

**Re.4 Thanks, We have rewrote the two puzzling sentences to be:**

**"From the origin of rocky desertification, its remediation should focus on vegetation restoration, (Wang et al., 2004). Consequently, the screening of plant species which can grow successfully in high-Ca environments in rocky desertification areas is an extremely critical step."**

P3L2: Change "Ca2+ and pectin in the cell walls of plants combine" into: "Ca2+ combine with pectin in the cell walls of plants"

P3L11: change" than cannot" into "not"

P4L3-L4 "Ji et al. (2009) revealed that the mean soil ECa was 3.61 g·kg-1 in the Puding, Huajing, Libo and Luodian

20    Counties of Guizhou Province, which is several times that of non-limestone areas in China." should be: "The mean soil ECa was 3.61 g·kg-1 in the Puding, Huajing, Libo and Luodian Counties of Guizhou Province, which is several times that of non-limestone areas in China (Ji et al. 2009).

P4L7-L19 "These results indicate that there are differences in soil Ca content between different areas and that there are differences between calcareous and non-calcareous plants in terms of Ca absorption, transport and storage and other

25    physiological processes. Collectively, these differences lead to different degrees of adaptability of plants to high Ca environments." Should be "There are variations in soil Ca content among different areas and differences between calcareous and non-calcareous plants in terms of Ca absorption, transport and storage and other physiological processes. These differences need to identify the variety of the plants to adapt with high Ca environments."

P4L10-L11 delete "there is a scarcity of extensive research into" should be" the mechanisms by which plants adapt to high

30    Ca conditions, particularly in karst areas and the Ca2+ dynamics of plants and soil are not well understood.

P4L14: delete "In order to", capitalize the "To"

P4L15 "we did the following:" should be "the following investigations were explored"

**Re.5 Thanks for the constructive suggestions! And we have corrected all of them.**

**3. Materials and methods**

P5: site description is too simple, should add more information regarding to the study, e.g. slope, soil pH, soil properties, and vegetation cover

**Re.6 Thanks, We have corrected it. In order to make site description more detailed, we added a table.**

**Table 1. Basic description for different grades of rocky desertification sites**

| Sample areas | Score of rocky desertification | Aspect | soil pH | Gradient (°) | Altitude (m) | Bedrock expose rate | Vegetation coverage | Disturbance regimes |
|---|---|---|---|---|---|---|---|---|
| LRD | 34(≤45) | South | 5.56 | 20° | 500 | 35% | 80% | Slight human disturbance, rarely grazing |
| MRD | 48(46~60) | Northeast | 5.57 | 18° | 500 | 57% | 75% | Abandoned farmland, no disturbance after abandoning cultivation |
| IRD | 67(61~75) | Southwest | 5.59 | 17° | 480 | 73% | 40% | Slight human disturbance, rarely grazing |

P5L4 title "Data collection" should be "Experimental design and data collection"

P5L6 delete "period. These four main indices"

P5L11: "We conducted a detailed survey of the three sample areas and collected samples in October 2016." Should be "The sample collection in these three sample areas were conducted in October 2016. "

**Re.7 Thanks, Very good suggestions! Done.**

P5L13: use "Within" to replace the "For"

P5L14: add were set up after "(upper, middle, and lower slope)"

P5L14-L15: add (3x4x3) were set up before the "for analysis", delete "We chose to study" and "the common plant species of the region, 15 and gathered plants using the whole plant harvest method. In each small quadrat, every kind of shrubs and herbs are collected." Should be "The common plant species of the region were gathered using the whole plant harvest method in each small quadrat as well as all shrubs and herbs are collected."

P5L17: "heated" should be "oven tried'

**Re.8 Thanks for your suggestions. We have corrected all of them.**

P5L18: add "and after" after the de-enzyme"

P5L18: not clear, "constant weight at 80oC, L 17 you mentioned 105oC, why?

Rewrite it

P5L18: delete "and bagged" and add "chemical" before the "analysis"

**Re.9 We add "and then" after the de-enzyme.**

**The 105°C is in order to de-enzyme, and the time does not need too long (only 15minutes). The 80°C is designed to make samples complete dehydration and it take a long time, but excessive temperature will carbonize the sample.**

**And this sentence should be: Plant samples were taken back to the laboratory, rinsed with distilled water before being oven tried at 105°C for 15 min to de-enzyme, and then dried to a constant weight at 80°C for about 480 minutes, crushed and passed through a 0.149 mm sieve, for later chemical analysis.**

P5L30: delete "Finally" add "were sampled"

**Re.12 Thanks, "Finally" was deleted, but "were sampled" cannot be added.**

**This sentence should be: "Soil TCa, ECa content and plant Ca content were measured using an Atomic Absorption Spectrophotometer (3510, Shanghai, China)."**

P6L5: "biennial herbs, while 'deciduous shrubs' included deciduous trees with a height less than 2 m or a ground diameter less than 3 cm." not clear, rewrite

**Re.13 In the small quadrats, there were very few biennial herbs, as a result, we gathered them to the 'Annual herbs'. We rewrite this sentence as:**

**The biennial herbs were gathered to the 'Annual herbs'. The deciduous trees with a height less than 2 m or a ground diameter less than 3 cm were gathered to the 'deciduous shrubs'.**

P6L8-L9: "One-way analysis of variance (ANOVA), Two-way ANOVA and Pearson correlation analysis ($\alpha = 0.05$) were used to analyze the Ca content of soil and plants within and between different grades of rocky desertification." Not clear, rewrite it

**Re.14 Sorry for the confusing, We have rewrote it as:**

**"We carried out two-way ANOVA for both species and soil for these 17 plants to determine differences in plant Ca content. One-way analysis of variance (ANOVA) was used to analyze the Ca content of soil and plants between different grades of rocky desertification. Pearson correlation analysis ($\alpha = 0.05$) was used to analyze the correlation between plant Ca and soil ECa content."**

**4. Results**

P6L13: add in soil after "The mean TCa content"

P6L14: Use "location" to replace "points"

P6L15: delete "Furthermore", and add "The", and to use "that" replacing "to be"

P6L17: Add "Ca content" after: "average", and use "the" to replace "with"

P6L21: Use "Total" to replace "The"

P7L1: use "." To replace ",", and then use "Compare to" to replace "while"

P7L3: delete "when compared across", to use "throughout"

P7L4: "aboveground" add "and belowground", and delete "or that of underground parts, there"

P7L5: Delete the whole sentence "Furthermore, the grades of rocky desertification had no obvious effect on the Ca content of the aboveground and underground parts of the plants generally (Fig. 2)."

P7L8: "The 41 plant species were identified in the 36 small quadrats; these plants were divided into different functional groups" should be "The 41 plant species were identified and were divided into different functional groups in the 36 small quadrats."

P7L8-L9: delete "For each functional group," add "The" before Ca

P7L9-L10: "Ca content between the aboveground and underground parts were significantly different (p< 0.05), and 10 the Ca content of the aboveground parts was higher than that of the underground parts (p<0.05)" should be " The Ca content of the aboveground parts significantly was higher than that of the belowground parts in each group (p<0.05)"

P7L15-L16: "In terms of life form functional groups, shrubs showed a significantly higher Ca content, both aboveground and underground than herbs (p<0.05)" should be "In life form functional groups, shrubs showed a significantly higher in Ca content than herbs in both aboveground and underground (p<0.05)"

P7L18: "The aboveground and underground Ca content of dicotyledons" should be"

The Ca content of dicotyledons in aboveground and belowground parts"

**Re.15 Thank you for these suggestions. We have corrected all of them.**

P7L21-23: "In terms of monocotyledons and dicotyledons, further analysis revealed no significant differences in the Ca content of the aboveground parts when compared between the different grades of rocky desertification; this was also true for the Ca content of the underground parts." Its not clear, rewrite it

**Re.16 Sorry for the confusing, We have rewrote it as:**

**To monocotyledons and dicotyledons, there were no significant differences in the plant Ca content of the aboveground parts among the different grades of rocky desertification; this was also true for the plant Ca content of the belowground parts.**

P7L23: Delete "The Ca content of both the aboveground and underground parts of monocotyledons was always low while those of dicotyledons were always high"

**Re.17 Thanks. Done !.**

P8L1-2 "The Ca content of dicotyledons was significantly higher than those of monocotyledons across" should be "The Ca content of dicotyledons in both of was significantly higher than those of monocotyledons in both aboveground and belowground parts throughout"

**Re.18 Yes. But we think this sentence should be: The Ca content of dicotyledons was significantly higher than those of monocotyledons in both aboveground and belowground parts throughout.**

P8L3: "For the 41 common plants collected, 17 plant species (which exist in each sample area) were widespread throughout the southwestern rocky desertification areas of Hunan." Should be "Within total 41 common plants species, 17 plant species were found in each sample plot and were widespread throughout the southwestern rocky desertification areas of Hunan."

P8L5: Delete "For each of t", capitalize "T"

P8L3: use were calculated replace "we calculated"

P8L5: Delete ". These plants were common species in the local area"

**Re.19 Thanks for the constructive suggestions! We have corrected them.**

P8L5-"We carried out two-way ANOVA for both species and soil for these 17 plants to determine differences in plant Ca content" should be moved to the data analysis part, not in the results part

**Re.20 Yes. We have moved it to the data analysis part. (See Re. 14).**

P8L6: Delete ". The soil was graded into three categories: LRD, MRD and IRD."

P8L7: Delete "df=16, F=11.277"

P8L8: Use "related among" to replace "significant among the different

P9L9: (df=2, F=2.299, p=0.117)

P8L9: "The" Ca not For "Ca", delete "differences"

P8L10: Use "among the species", delete "(df=16, F=8.543, p<0.01)", and delete "but also among the different" and it throughout all the" , and delete "df=2, F=4.104,"

**Re.21 Thanks. We have corrected them.**

P8L12-13: "The correlation between plant Ca content and soil ECa content reflects what extent soil Ca content influences

plant Ca content, and may also reflect how different plants respond to differences in soil ECa content" this sentence should not in results part, may be in discussion section.

**Re.22 Yes. We have put it in the discussion section (4.2).**

5    P8l13: Too many "For this" and "In terms of", delete them.

**Re.23 Agree. And we have deleted them.**

P8L15: "which indicated that Sanguisorba officinalis was affected greatly by soil ECa conten" should be not in the results section.

10   P8L17: "indicating that the underground parts of these species were also greatly affected by soil ECa content." should be not in the results section.

P8L19" which indicated that the aboveground parts of Themeda japonica was also greatly affected by soil ECa 20 content" should be not in the results section

**Re.24 Thanks. We have put them in the discussion section (4.2).**

P9L2: Delete "kinds of"

P9L3: "and were also the representative species that are able to adapt to a high Ca soil environment." How do you know it? Suggest to delete it

**Re.25 Agree. And done!.**

P9L6-9: "The capacity of these plants which are able to adapt to high Ca soil environments can be reflected by two indicators: (i) the correlation between Ca content in the aboveground parts of the plants and soil ECa content; (ii) the species differences in terms of the Ca content of the aboveground parts of plants. Thus, based on the above two indicators, we classified these plants into the following groups: Ca-indifferent plants, high-Ca plants and low-Ca plants (Ji et al., 2009)."

25   This should be moved to the discussion section. Results just present your results, no explanation and citation.

**Re.26 Thanks. we have put it in the discussion section (4.3).**

P9L10: The definition "Ca-indifferent plants" is it correct?

**Re.27 Yes. It is correct.**

P9L12:" The Ca content of these plants increased or decreased correspondingly with increases or reductions in soil ECa content, but plant growth was not affected by such changes." Not clear, rewrite it

**Re.28 Yes. This sentence was rewrote and was moved to 4.3. This sentence was:**

**In both high-Ca and low-Ca soil environments, the Ca-indifferent plants can survival normally. And the Ca content of them changes correspondingly with the change of soil ECa content.**

5   P9L17: "High-Ca plants", refer it "high Ca demand plants

    **Re.29 Yes. But High-Ca plants, Low-Ca plants and Ca-indifferent plants were used as terminology. See Ji, F. T., 2009.**

P9L20: "Moreover, the physiological activities of these plants had a higher demand for Ca and may have a strong ability to enrich soil Ca." should be in the discussion part

10   **Re.30 Yes. We have put it in the discussion section (4.3)**

P9L21: "Low-Ca plants" should be Low Ca demand plants

    **Re.31 Yes. The same as Re. 29.**

15   P9L23: Why do you use "19g/kg" as the boundary to determine the low or high Ca demand plants?

    **Re.32 Mainly according to the data of calcium content of plant (from 0.42 to 41.79 g·kg$^{-1}$ ) and refer to the relevant references( See Ji, F. T., 2009).**

P10L2-5: the whole paragraph should belong to the discussion, not in the results. Again, the results should just present your

20  results, do not need any explanation in this part, any explanation and citation should be in the discussion section.

    **Re.33 Thanks. We have put it in the discussion section (4.3)**

**5. Discussion**

P10L9: delete "The aboveground parts of plants had a higher Ca content than the underground 10 parts, although"

25  P10L10: Capitalize T (The), delete when compared

P10L17: period after the (2014), and then separate the paragraph

    **Re.34 Thanks .**

**For the first two questions, we deleted "The aboveground parts of plants had a higher Ca content than the underground parts, although there was no significant difference in plant Ca content when compared between**

**aboveground or underground parts (p>0.05) across the different grades of rocky desertification. This indicated that the grade of rocky desertification had no obvious effect on the Ca content of the aboveground and underground parts of the plants studied herein.**

**For the third question, we did not separate the paragraph**

P10L18: What is the ABC soil?

**Re.35 Thanks. the ABC is a shorthand (full name: acid buffering capacity).**

P10L18: "Tanikawa et al. (2017) revealed that concentrations of TCa and ECa were also low at the deeper horizons in the low-acid buffering capacity (ABC) soils, and pointed to differences in both organic layer thickness and soil chemistry as a reason for affecting Ca accumulation of low- and 20 high-ABC stands" is unclear, rewrite it

**Re.36 We have rewrote it. This sentence should be: "Tanikawa et al. (2017) revealed that concentrations of TCa and ECa were also low at the deeper horizons in the low-acid buffering capacity (ABC) soils, and differences in both organic layer thickness and soil chemistry could be a reason affecting Ca accumulation of low- and high-ABC stands".**

P11L1: Add "compared to the aboveground and belowground Ca content in our study," before the "The", and then use lowercase of "t"
P11L3: "," should be "."

**Re.37 Thanks . We did not added "compared to the aboveground and belowground Ca content in our study,". The next sentence should be: the maximum and minimum Ca content of plant aboveground parts were 41.79 g·kg$^{-1}$ and 2.15 g·kg$^{-1}$ respectively, and the maximum and minimum Ca content of plant belowground parts were 40.14 g·kg$^{-1}$ and 0.42 g·kg$^{-1}$ respectively. The maximum Ca content of plant (41.79 g·kg$^{-1}$) was found in the leaves which is lower than the Ca content of calcareous plants leaves with the maximum value of 85.13 g·kg$^{-1}$ by Luo et al. (2014).**

P11L1-4: "The maximum and minimum calcium content of plant aboveground parts were 41.79 g·kg-1 and 2.15 g·kg-1 respectively, and the maximum and minimum calcium content of plant underground parts were 40.14 g·kg-1 and 0.42 g·kg-1 respectively, Which is lower than the calcium content of calcareous plants leaves (maximum 85 .13 g·kg-1 ,minimum 6.26

g·kg-1) by Luo et al. (2014)." Aboveground includes leaves and branches, how do you compared with leaves only? Ca presents the Calcium, should keep the constant in the manuscript.

**Re.37 Sorry for the confusing. This sentence should be:**

**The maximum and minimum Ca content of plant aboveground parts were 41.79 g·kg⁻¹ and 2.15 g·kg⁻¹ respectively,**

5     **and the maximum and minimum Ca content of plant belowground parts were 40.14 g·kg⁻¹ and 0.42 g·kg⁻¹**

**respectively. The maximum Ca content of plant (41.79 g·kg⁻¹) was found in the leaves which was lower than the Ca**

**content of calcareous plants leaves with the maximum value of 85.13 g·kg⁻¹ by Luo et al. (2014).**

P11L9: The beginning of the paragraph should present your research results pattern first, and then discuss and explain it.

10   P11L11: Use "had a" to replace "was extremely". Use "." and delete "and" to separate the sentence. The sentence "our results showed several plants (*Sanguisorba officinalis, Dendranthema indicum, Castanea henryi and Themeda japonica* ) and soil Eca content was a positive correlation, but most plant calcium content and soil ECa content was not relevant." Should be "Our results showed that most plants had no correlation relationship between soil ECa and plant Ca excepting several plant (*Sanguisorba officinalis, Dendranthema indicum, Castanea henryi and Themeda japonica* ) had a positive

15   correlation between soil Eca and plant Ca content."

**Re.38 Thanks . We have corrected it.**

P11L14: what are "species-related factors,"? Do you mean plant species physiological factors?

**Re.39 No. 'species-related factors' refers to 'species factors'.**

P11L15-16: "was in accordance with data reported previously by Ji et al. (2009)." should be "was supported with data reported by Ji et al. (2009)."

**Re.40 Yes and thanks. Done !**

25   P11L17: "Since the transport of Ca was mainly one-way (upward), this result indicated that nitrogen-fixing plants were the most efficient in terms of the upward transport of Ca, and that Ca was mainly concentrated in the aboveground parts of the plant; these findings were not consistent with those of Ji et al. (2009)." Is not clear, rewrite it.

**Re.41 Sorry. We have rewrote it.**

**The Ca content in the aboveground parts of nitrogen-fixing plants was significantly higher than that of the**

30   **belowground parts. And this result indicated that nitrogen-fixing plants were the most efficient in the Ca upward**

**transport, since the transport of Ca was mainly upward; which was not the same with those of Ji et al. (2009). Ji et al. (2009) revealed that ......**

P11L19: delete "In the paper"

P11L20: "." after Ca, The sentence" and studied only three types of plants (pteridophytes, dicotyledons, monocotyledons) that did not include nitrogen-fixing plants, which may be the reason for the inconsistency of this previous data with our current findings." Should be" They used three types of plants (pteridophytes, dicotyledons, monocotyledons) exclude nitrogen-fixing plants in their study, which may have a conflicting result compared with our current findings."

P11L22-P12L1: delete "in terms of", the sentence "In terms of the Ca content of monocotyledons, we found significant differences (p<0.01) between the aboveground and underground parts, but the study by Ji et al. (2009) revealed that these differences were not significant. This may be because most of the monocotyledons collected were low-Ca plants." should be" We found significant differences (p<0.01) between the aboveground and belowground parts in Ca content of monocotyledons in our study. However, Ji et al. (2009) revealed that no significant differences between the aboveground and belowground parts in Ca content of monocotyledons. This phenomenon may contribute the most of the monocotyledons sample plants were low-Ca demand plants."

**Re.42 Thanks. We have corrected them.**

P12L2-3: "Owing to the fact that the aboveground parts of low-Ca plants maintain a lower Ca content for different grades of rocky desertification, a significant difference was found between the aboveground and underground parts in monocotyledons. In addition, the Ca content of monocotyledons was lower than that reported for monocotyledons (Ji et al., 2009)," is not clear, rewrite it

**Re.43 Thanks. We have rewrote it as: "A significant difference was found between the aboveground and belowground parts in monocotyledons, which may be because low-Ca plants maintain a lower Ca content in different grades of rocky desertification. In addition, the Ca content of monocotyledons was lower than that reported for monocotyledons (Ji et al., 2009)".**

P12L7-8: "Over the past decade, progress has been made in identifying the cellular compartments (e.g., endoplasmic reticulum, chloroplasts and mitochondria) that regulate Ca balance and signal transduction in plants (Müller et al., 2015). " may move to the introduction section.

**Re.44 Yes, We have put it in the introduction section.**

P12L8-P13L14: again, authors should present the results pattern at the beginning of the discussion to explain your results.

This paragraph should be rewritten. Lots of "in terms of" showed in the manuscript, delete them. In this paragraph, I did not see any results at the beginning of the discussion. The discussion is used to explain the results

**Re.45 Thanks. We have deleted it, and added some content.**

5    P13L15-22: suggest deleting the paragraph because it does not make sense in your discussion, as well as so many times to cite the literature Ji et al (2009)

**Re.46 Thanks . We have deleted it.**

**6. Conclusions**

10   P13L5 "Conclusions"

P1323-P14L1: delete "followed by" add "and" before "LRD"

P14L1-2: delete the sentence" Significant differences were detected for both soil ECa and TCa content when compared between the rocky side and non-rocky side of each grade of rocky desertification areas. "

P14L3: add "sites" after "studied", delete "Furthermore"

15   P14L5: Delete "(p<0.05)

L14L6;  Delete "the" Ca

**Re.47 Thanks. We have deleted them.**

P14L6: Ca-indifferent" is correct?

20   **Re.48 Yes. Thanks.**

P14L7: "," after "Themeda japonica", delete "For these plants," and put had

P14L8: "High-Ca plants included Pyracantha fortuneana," should be" High-Ca plants in our study were Pyracantha fortuneana,

25   P14L10: delete "In this case",  the sentence "the aboveground parts of these plant were able to maintain a higher Ca content under conditions of variable soil ECa content. "should be" the aboveground parts of these plant were able to absorb a high Ca content from various of ECa content soils.

**Re.49 Thanks. We have corrected them.**

1. General comments

The authors' manuscript "Calcium content and high calcium adaptation of plants in karst areas of southwestern Hunan, China" introduced an investigation and an analysis of the relationships between the degree of rocky desertification and calcium content in soil and plant. The author's results are interesting. These results can be seen as a valuable reference, which could be helpful in related research works to screen plant species for vegetation restoration in karst areas of China. As for the study itself, this paper is worthy to be published in the journal "Biogeosciences". However, this manuscript needs a major revision before publication, especially revisions in paper structure and writing quality.

**Re.1 Thanks for the positive comments.**

2. Specific comments

Because there is no line number in the manuscript that I downloaded, I suggest the authors put numbers for each line in the revised version if the manuscript goes to the next stage. Here I provide my comments for the sections.

(1) For "Introduction"

In the section of Introduction, I cannot find sufficient reviews or introductions of existing studies on the issue addressed by the authors' study. There are only two existing studies (Zhang 2005; Ji et al., 2009) mentioned in this section. The study background need be introduced more in this section. I wonder if the authors can provide a brief review for this issue including some important related studies reported for other countries. Readers may want to know whether the authors' hypothesis, "the dynamics of Ca content is significantly affected by the grade of rocky desertification", is supported by more studies or not.

The second, third, and fourth paragraphs are lengthy in describing the knowledge of plant physiology. I suggest shortening them to several sentences for outlining some key processes.

**Re.2 Thanks for the suggestion. We have added 3 references in this section. To the second, third, and fourth paragraphs. We have restructured and shorted these paragraphs.**

(2) For "Results"

The major problem is that there are lots of explanations and analyses in this section, especially in section 3.2.2 and 3.4. Of course, for a better understanding, it may be ok to arrange a few explanations in "Results". But any analysis should not appear in the "Results". Otherwise, this section can be "Results and discussions".

**Re.3 Thanks, this is a very good suggestion. We have moved all sentences with explanations to the discussion section (4.3). In addition, we also modified the section 3.3.**

(3) For "Discussions"

Overall, there are still too much knowledge descriptions of plant physiology. I think that only the necessary knowledge should be mentioned corresponding to the new findings, instead of a detailed introduction of knowledge. I would rather see that the discussions focus on the main points and your hypothesis, or analyze your main work, for example, the three parts of your work: (i) to measure the soil Ca contents; (ii) to compare between above- and below-ground parts of plants; and (iii) to analyze correlation between Ca in plants and soils.

**Re.4 Thank you for your constructive suggestions. And done accordingly!**

For section 4.1:

According to the text, the authors may discuss their works (i) and (ii) in section 4.1. However, section 4.1 seems not well organized. Actually, the results have already shown the dynamics of soil Ca, and the difference between above- and below-ground parts of plants. I hope the section 4.1 can summarize these two works clearly, and can indicate some new findings. In addition, for readability, readers may need to know percentages of your measurements to the values reported by other studies. I also suggest the authors consider changing the title from "Dynamics of Ca content in plants and soil" to "Dynamics and ranges of Ca content in plants and soil" since most parts in this section are talking about the "ranges".

**Re.5 Yes. We have changed the 4.1 title to "Dynamics and ranges of Ca content in plants and soil". The contents have also been reorganized, and the content was divided into two paragraphs. The first and second paragraphs discussed soil Ca content and plant Ca content, respectively. Thanks.**

For section 4.2:

The authors' work (iii) was discussed in this section. It seems to me, the content needs to be reorganized very logically. Section 4.2 lists some results and other researchers' conclusions, however, the logical relationships between those results and conclusions are not clear. For example, the first sentence states "The $Ca^{2+}$ content in plant cells was proportional to soil $Ca^{2+}$". Then what parts of plants are you talking about? Aboveground, below-ground, or whole tree? The second sentence states "Calcium-rich soils caused cells to absorb more calcium than the 10 cells themselves require (White and Broadley, 2003)". Then is this cited sentence for supporting the first sentence, just explaining the cause, or conducting the third sentence? A conjunction word seems necessary.

The third sentence states "Zou et al. (2010) showed that soil ECa content and leaf calcium content [were] extremely significant positive correlation". Is this conclusion cited for comparing your results? If so, it would be better to indicate the plant name(s). Your next statement is "our results showed several plants (......) and soil Eca content was a positive correlation, but most plant calcium content and soil ECa content was not relevant". Zou et al. focused on the leaf, but, do you focus on whole tree? The results and comparisons need to be explained more clearly and logically to avoid reader's confusion.

These above writing issues are raised just for example. The whole section needs to be rewritten for better readability

**Re.6 Thanks. We have rewrote the section 4.2.**

**(1) To the first sentence, we are sorry for the confusing sentences. And it was rewrote.**

**(2) The second sentence is in order to compare to our results (the next sentence). We have added a conjunction "but", and put the next sentence (our results) in front of this sentence. Furthermore, analyze the reasons that leading to different results.**

**(3) To the third sentence, this conclusion is for comparing our results, the plant is "Tobacco", and we have added the plant name in the manuscript. We analyzed the correlation between plant and soil calcium content, and our plant was divided into aboveground and underground parts (see Table. 3).**

**In addition to the three sentences mentioned above, other problems have also been revised.**

For section 4.3:

This section discusses the most important scientific issue (High calcium adaptation of plants) addressed in the study. The solution of this issue may provide useful guidance on vegetation restoration. However, basic knowledge descriptions take up lot of space. I had liked to see the discussions on: (a) Based on study results for the 17 selected species, what are the primary characteristics for each of the three categories (Ca-indifferent plants, high-Ca plants and low-Ca plants)? (b) What should the screening of plant species notice in the vegetation restoration? (c) What are the application prospects in solving the problem of land degradation using the authors' results in karst areas? (d) Is there any unsolved issue, related with this study and remained for further research?

Again, results are interesting and helpful for associated studies. I suggest accepting this manuscript after a major revision. The writing quality should be improved, including a spelling check. As I am not a native English speaker, I will not suggest more regarding language. Good luck!

**Re.7 Thanks you for your advice, and we have deleted lots of basic knowledge descriptions, only a small part reserved.**

**Re. to the question (a): We have added the primary characteristics of three categories in 3rd paragraph.**

**Re. to the question (b): We added the corresponding content in the fourth paragraph.**

**Re. to the question (c): Our findings not only have important guiding significance for solving the problem of rocky desertification in China, but also provide species screening ideas for the rocky desertification ecosystem restoration in other areas all over the world. And we have added this content in the forth paragraph.**

**Re. to the question (d): Thanks. It is necessary to further explore other nutrient element in soil during vegetation restoration, and long-term positioning observation is crucial for the study of this issue. We added this new content in the fourth paragraph.**

5 ## *Part 2:* A list of all relevant changes made in the manuscript:

1. P1L13: Removes the sentence "However, the Ca2+ dynamics of plants and soil are not well understood" into the P5 L14.
2. P1L16: deleted the" s" from the "samples"
3. P1L16: changed "slop" into "slope"
10 4. P2L2: changed "can provide" into "provides"
5. P2L8-10: inserted the "Rocky desertification is an extreme form of land degradation in karst areas, and 10 has become a major social problem in terms of China's economic and social development (Sheng et al., 2015)" after "……in China are mainly distributed in southwestern areas.", and deleted P2 L11-102 this sentence "Rocky desertification is an extreme form of land degradation in karst areas, and 10 has become a major social problem in terms of China's economic and
15 social development (Sheng et al., 2015)"
6. P2L10: deleted "Of course"
7. P2L10-11: changed the sentence "the severity of rocky desertification in Hunan Province was ranked fourth (Li et al., 2016)" into "The severity of rocky desertification was ranked in fourth in Hunan Province of China (Li et al., 2016)"
8. P2L12-14: changed the "The restoration and reconstruction of rocky desertification ecosystems has become the
20 immediate focus of agro-forestry production environment improvements, regional economic development and helping to support people out of poverty (Jing et al., 2016)" into "The restoration and reconstruction of rocky desertification ecosystems have become the urgent environment improvements, regional economic development by using agroforestry system and helping to support people out of poverty (Jing et al., 2016)
9. P2L16: inserted "with" between "Soil" and "high Ca"
25 10. P2L17-20: "From the origin of rocky desertification, the restoration of vegetation is key to the process of remediation (Wang et al., 2004). Consequently, the screening of plants which can grow successfully in high- calcium environments is an extremely critical step." was modified to "From the origin of rocky desertification, its remediation should focus on vegetation restoration, (Wang et al., 2004). Consequently, the screening of plant species which can grow successfully in high-Ca environments in rocky desertification areas is an extremely critical step."
30 11. P2L21-23: "$Ca^{2+}$ is one of the most essential nutrients needed for the regulation of plant growth and also plays a central role in helping plants overcome environmental stress (Hepler, 2005)" was modified to "$Ca^{2+}$ is one of the most essential

nutrients needed for the regulation of plant growth and is also plant signal sensor (second messenger) under conditions of environment stress (Poovaiah and Reddy, 1993; Hepler, 2005; Hong-Bo and Ming, 2008; Batistič and Kudla, 2012)"

12. P3L1-4: deleted "A low cytosolic $Ca^{2+}$ concentration is crucial for appropriate cell signaling (Müller et al., 2015). At the same time, $Ca^{2+}$ is a versatile plant signal sensor under conditions of soil water stress (Hong-Bo and Ming, 2008). In addition, $Ca^{2+}$ as a second messenger in the process of cell signal transduction, which plays a key regulatory role in how plants respond to environmental changes (Poovaiah and Reddy, 1993; Batistič and Kudla, 2012)."

13. P3L6-7: deleted "However, high Cacalcium stress can exert influence over the photosynthetic and growth rate of plants (Ji et al., 2009; Hui et al., 2003)"

14. P3L7-8: changed "$Ca^{2+}$ and pectin in the cell walls of plants combine to form pectin Calcium" into: "$Ca^{2+}$ combine with pectin in the cell walls of plants to form pectin Ca"

15. P3L11: inserted "Over the past decades, the progress has been made in identifying the cellular compartments (e.g., endoplasmic reticulum, chloroplasts and mitochondria) that regulate Ca balance and signal transduction in plants (Müller et al., 2015)." after "Mechanisms of plant defense to high soil $Ca^{2+}$ concentrations:"

16. P3L14: inserted "; Larkindale and Knight, 2002" before "Borer et al., 2012"

17. P3L14-15: changed "Plants can be adapted to high salt environments by activating the $Ca^{2+}$ signal transduction pathway (Bressan et al., 1998)" into "Plants can be adapted to high salt, drought and high temperature environments by activating the $Ca^{2+}$ signal transduction pathway (Bressan et al., 1998)"

18. P3L16-19: deleted "Drought is a common environmental stress factor in rocky desertification areas, and high temperatures enhance the degree of heat damage, causing oxidative damage to the cell membrane. However, if the Ca2+ concentration of plants can be increased, this process can be effectively inhibited, thereby preventing or reducing heat damage (Larkindale and Knight, 2002). A fine regulatory mechanism exists in t"

P3L19: changed "the plant" into "The plant", and deleted "that can"

P3L21: deleted ", which play a key role in the $Ca^{2+}$ transport system in plants"

19. P3L22-24: deleted "The $Ca^{2+}$ transport system ($Ca^{2+}$ channel, $Ca^{2+}$ / $H^+$ reverse transport carrier and $Ca^{2+}$-ATPase) plays an important role in the uptake, transport and distribution of Ca in plants (White and Broadley, 2003)."

20. P4L3-5: deleted "The Ca content of plants usually lies between 0.1 % and 5.0 %, and mostly exists in cell walls and vacuoles - in the form of pectin combination morphology and insoluble organic and inorganic Ca salts (Kinzel, 1989)."

21. P4L11-21: inserted "Plants adaptation to high Ca soil environment: Some plants fix excess $Ca^{2+}$ by forming calcified deposits in root tissue in order to limit the upward transport of $Ca^{2+}$ (Musetti and Favali, 2003). In addition, Ca oxalate crystals in the plant's crystal cells play a role in regulating plant Ca content (Ilarslan et al., 2001; Pennisi and McConnell, 2001; Volk et al., 2002). In a high Ca environments, some plants will form Ca oxalate crystal cells in order to fix excess $Ca^{2+}$ (Moore et al., 2002). Furthermore, an active Ca efflux system plays an important role in the adaptation of plants to high Ca environments (Bose et al., 2011): Excess $Ca^{2+}$ in plants is exported from mature leaves to the outside, thereby maintaining a lower concentration of leaf Ca (Musetti and Favali, 2003). The regulation of internal Ca storage predominantly depends on plasma membrane Ca transport and intracellular Ca storage; collectively these processes can regulate the intracellular $Ca^{2+}$ concentration to a lower level (Bowler and Fluhr, 2000). Plants that adapt to high Ca

environments promote excess $Ca^{2+}$ flow through the cytoplasm or store $Ca^{2+}$ in vacuoles via the cytoplasmic $Ca^{2+}$ outflow and influx system (Shang et al., 2003; Hetherington and Brownlee, 2004; Wang et al., 2006)."

22. P4L22-23: changed "Ji et al. (2009) revealed that the mean soil ECa was 3.61 g·kg$^{-1}$ in the Puding, Huajing, Libo and Luodian Counties of Guizhou Province, which is several times that of non-limestone areas in China." into "The mean soil ECa was 3.61 g·kg$^{-1}$ in the Puding, Huajing, Libo and Luodian Counties of Guizhou Province, which is several times that of non-limestone areas in China (Ji et al. 2009)."

23. P4L23-24: inserted "Wang et al. (2011) found that plant rhizosphere soil TCa content in calcareous soil area ware above 14.0 mg·g-1." before "Zhang (2005).….."

24. P5L3-7: inserted "Luo et al. (2013) showed that $Ca^{2+}$ concentration affected plant photosynthesis. When the daily net photosynthetic rate of *Cyrtogonellum Ching* and *Diplazium pinfaense Ching* reached the highest value, the concentrations of $Ca^{2+}$ were 30 mmol·L-1 and 4 mmol·L-1, respectively. Qi et al. (2013) found that a significant difference in calcium content among *Primulina* species from different soil types, with high average calcium content (2,285.6 mg/kg) in *Primulina* from calcareous soil relative to low levels present in *Primulina* from both acid soil (1,379.3 mg/kg) and Danxia red soil (1,329.1 mg/kg)." after " ……,but inhibit growth in *Rhododendron decorum*."

25. P5L7-13: changed "These results indicate that there are differences in soil Ca content between different areas and that there are differences between calcareous and non-calcareous plants in terms of Ca absorption, transport and storage and other physiological processes. Collectively, these differences lead to different degrees of adaptability of plants to high Ca environments." into "There are variations in soil Ca content among different areas and differences between calcareous and non-calcareous plants in terms of Ca absorption, transport and storage and other physiological processes. These differences need to identify the variety of the plants to adapt with high Ca environments."

26. P5L13-15: changed "However, to date, there is a scarcity of extensive research into the mechanisms by which plants adapt to high Ca conditions, particularly in karst areas." into "However, to data, the mechanisms by which plants adapt to high Ca conditions, particularly in karst areas and the $Ca^{2+}$ dynamics of plants and soil are not well understood."

27. P5L18-19: deleted "In order to", capitalize the "To". And changed "we did the following:" into "the following investigations were explored"

28. P6L2: changed "was" into "is"

29. P6L7: inserted "(see Table. 1)" after "…….is poor", and add a "Table 1. Basic description for different grades of rocky desertification sites" in P24.

30. P6L8: changed title "Data collection" into "Experimental design and data collection"

31. P6L10: deleted ". These four main indices"

32. P6L13-14: changed "range of characteristics and data relating to the surrounding environment" into "environmental factors"

33. P6L15-16: changed "We conducted a detailed survey of the three sample areas and collected samples in October 2016." into "The sample collection in these three sample areas were conducted in October 2016.

34. P6L18: use "Within" to replace the "For", deleted "we assigned", and changed "2×2" into "(2×2)"

35. P6L19: add "were set up" after "(upper, middle, and lower slope)"

36. P6L19-L15: add "(3x4x3)" before the "for analysis", deleted "We chose to study"

37. P6L19-L21: changed "the common plant species of the region, and gathered plants using the whole plant harvest method. In each small quadrat, every kind of shrubs and herbs are collected." into "The common plant species of the region were gathered using the whole plant harvest method in each small quadrat as well as all shrubs and herbs were collected."

38. P6L22: deleted "parts" before "."

39. P7L1: changed "heated" into "oven tried', and add "and then" after "the de-enzyme,"

    P7L1: inserted "about 480 minutes" after "at 80°C", and deleted "for". Deleted "and bagged" and add "chemical" before the "analysis"

40. P7L3: deleted "Finally", and changed "soil" into "Soil"

41. P7L9-13: changed "'Annual herbs' included both annual herbs and biennial herbs, while 'deciduous shrubs' included deciduous trees with a height less than 2 m or a ground diameter less than 3 cm. The aboveground part of plants included branches and leaves, while the underground part included roots." into "The biennial herbs were gathered to the 'Annual herbs'. The deciduous trees with a height less than 2 m or a ground diameter less than 3 cm were gathered to the 'deciduous shrubs'. Branches and leaves were treated together as aboveground part, while the belowground part only included roots."

42. P7L13-17: changed "One-way analysis of variance (ANOVA), Two-way ANOVA and Pearson correlation analysis (α = 0.05) were used to analyze the Ca content of soil and plants within and between different grades of rocky desertification." into "We carried out two-way ANOVA for both species and soil for these 17 plants to determine differences in plant Ca content. One-way analysis of variance (ANOVA) was used to analyze the Ca content of soil and plants between different grades of rocky desertification. Pearson correlation analysis (α = 0.05) was used to analyze the correlation between plant Ca and soil ECa content."

43. P7L21: add "in soil" after "The mean TCa content"

44. P7L22: changed "points" into "location"

45. P8L1: deleted "Furthermore", add "The", and changed "to be" into "that"

46. P8L3-4: add "Ca content" after "average", and use "the" to replace "with"

47. P8L8: changed "The 41" into "Total 41"

48. P8L9: use "." To replace ",", and then use "Compare to" to replace "while"

49. P8L11: deleted "when compared across", changed into "throughout"

50. P8L12-13: add "and belowground" after "aboveground", and deleted "or that of underground parts, there"

51. P8L14-15: deleted "Furthermore, the grades of rocky desertification had no obvious effect on the Ca content of the aboveground and underground parts of the plants generally."

52. P8L17-18: changed "The 41plant species were identified in the 36 small quadrats; these plants were divided into different functional groups" into "The 41 plant species were identified and were divided into different functional groups in the 36 small quadrats."

53. P8L18: deleted "For each functional group,"

54. P8L18-20: changed "Ca content between the aboveground and underground parts were significantly different (p< 0.05), and the Ca content of the aboveground parts was higher than that of the underground parts (p<0.05) (Fig.3)." into "The Ca content of the aboveground parts was significantly higher than that of the belowground parts in each group (p<0.05)."

55. P9L3-5: changed "In terms of life form functional groups, shrubs showed a significantly higher Ca content, both aboveground and underground than herbs (p<0.05)" into "In life form functional groups, shrubs showed a significantly higher in Ca content than herbs in both aboveground and underground (p<0.05)"

56. P9L7- 8: changed "The aboveground and underground Ca content of dicotyledons" into "The Ca content of dicotyledons in aboveground and belowground parts"

57. P9L9: add "(Fig. 3)" after "(9.63 g·kg$^{-1}$ and 4.79 g·kg$^{-1}$, respectively; p<0.05)"

58. P9L10-12: changed "In terms of monocotyledons and dicotyledons, further analysis revealed no significant differences in the Ca content of the aboveground parts when compared between the different grades of rocky desertification; this was also true for the Ca content of the underground parts." into "To monocotyledons and dicotyledons, there were no significant differences in the plant Ca content of the aboveground parts among the different grades of rocky desertification; this was also true for the plant Ca content of the belowground parts."

59. P9L12-13: deleted "The Ca content of both the aboveground and underground parts of monocotyledons was always low while those of dicotyledons were always high"

60. P9L13-15: changed "The Ca content of dicotyledons was significantly higher than those of monocotyledons across" into "The Ca content of dicotyledons was significantly higher than those of monocotyledons in both aboveground and belowground parts throughout"

61. P9L16-17: changed "For the 41 common plants collected, 17 plant species (which exist in each sample area) were widespread throughout the southwestern rocky desertification areas of Hunan." into "Within total 41 common plants species, 17 plant species were found in each sample plot and were widespread throughout the southwestern rocky desertification areas of Hunan."

62. P9L17: deleted "For each of t", and add capitalize "T"

63. P9L18: use "were calculated" replace ",we calculated", and changed "Table.2" into "Table.3"

64. P9L18: deleted ". These plants were common species in the local area"

65. P9L18-20: deleted "We carried out two-way ANOVA for both species and soil for these 17 plants to determine differences in plant Ca content. The soil was graded into three categories: LRD, MRD and IRD."

66. P9L21: deleted "df=16, F=11.277"

67. P9L21-22: Use "related among" to replace "significant among the different"

68. P9L22: deleted (df=2, F=2.299, p=0.117)

69. P9L22: changed "For Ca" into "The Ca"

70. P9L23: deleted ", differences", and add "difference" before "not only"

71. P9L23-P10L2: changed "in terms of plant species" into "among the species", deleted "(df=16, F=8.543, p<0.01)", and deleted "but also among the different" and add "it throughout all the" before "grades of rocky desertification", and

deleted "df=2, F=4.104,"

72. P10L4-5: deleted "The correlation between plant Ca content and soil ECa content reflects what extent soil Ca content influences plant Ca content, and may also reflect how different plants respond to differences in soil ECa content", and changed "For these17" into "These 17"

73. P10L7: deleted ", which indicated that *Sanguisorba officinalis* was affected greatly by soil ECa conten"

74. P10L8: add "($p<0.01$)" after "*Castanea henryi*"

75. P10L19-10: deleted ", indicating that the underground parts of these species were also greatly affected by soil ECa content"

76. P10L11-12" deleted ", which indicated that the aboveground parts of *Themeda japonica* was also greatly affected by soil ECa 20 content", and changed "For the other" into "The other"

77. P10L13: changed "(Table. 3)" into "(Table. 4)"

78. P10L15: deleted "kinds of"

79. P10L15-16: deleted ", and were also the representative species that are able to adapt to a high Ca soil environment."

80. P10L18-22: deleted "(Grubb and Edwards, 1982). The capacity of these plants which are able to adapt to high Ca soil environments can be reflected by two indicators: (i) the correlation between Ca content in the aboveground parts of the plants and soil ECa content; (ii) the species differences in terms of the Ca content of the aboveground parts of plants. Thus, based on the above two indicators, we classified these plants into the following groups: Ca-indifferent plants, high-Ca plants and low-Ca plants (Ji et al., 2009)."

81. P11L3-4: deleted "The Ca content of these plants increased or decreased correspondingly with increases or reductions in soil ECa content, but plant growth was not affected by such changes."

82. P11L11-12: deleted "Moreover, the physiological activities of these plants had a higher demand for Ca and may have a strong ability to enrich soil Ca."

83. P11L14-16: deleted ". In addition, the physiological activities of these plants had a lower demand for Ca and could alleviate high Ca stress by inhibiting the absorption of Ca through the root system and its upward transport"

84. P11L16: changed "(Table. 4)" Into "(Table. 5)"

85. P11L17-20: deleted "Finally, the different plant functional groups revealed the differences in Ca content (Fig. 2). In some cases, even within the same plant, there was an inconsistent correlation between Ca content in the aboveground and belowground parts and the soil ECa content. Collectively, these findings showed that not all plants adapted to soil high Ca environments in the same way, and that they exhibited a variety of adaptive mechanisms."

86. P12L3-7: deleted "The aboveground parts of plants had a higher Ca content than the…… the plants studied herein."

87. P12L14: deleted "pointed to", and add "could be" after "and soil chemistry", deleted "as"

88. P12L18-20: add "There was no significant difference in plant Ca content between aboveground or belowground parts (p>0.05) across the different grades of rocky desertification. This indicated that the grade of rocky desertification had no obvious effect on the Ca content of the aboveground and belowground parts of the plants studied herein. But" after "(the main chemical composition: CaCO3) (Ji et al., 2009)."

89. P12L20-21: changed "The average calcium content of aboveground parts of plants was 19.67 g·k$^{g-1}$, which was lower

than that of Hunan flue-cured tobacco (21.93 g·kg$^{-1}$) (Xu et al., 2007)." into "the average Ca content of aboveground parts of plants (19.67 g·kg$^{-1}$) was lower than that of Hunan flue-cured tobacco (21.93 g·kg$^{-1}$) ( Xu et al., 2007)."

90. P12L22-P13L2: changed "The maximum and minimum calcium content of plant aboveground parts were 41.79 g·kg$^{-1}$ and 2.15 g·kg$^{-1}$ respectively, and the maximum and minimum calcium content of plant underground parts were 40.14 g·kg-1 and 0.42 g·kg$^{-1}$ respectively, which is lower than the calcium content of calcareous plants leaves (maximum 85.13 g·kg$^{-1}$,minimum 6.26 g·kg$^{-1}$) by Luo et al. (2014)." into "The maximum and minimum Ca content of plant aboveground parts were 41.79 g·kg$^{-1}$ and 2.15 g·kg$^{-1}$ respectively, and the maximum and minimum Ca content of plant belowground parts were 40.14 g·kg$^{-1}$ and 0.42 g·kg$^{-1}$ respectively. The maximum Ca content of plant (41.79 g·kg$^{-1}$) was found in the leaves which was lower than the Ca content of calcareous plants leaves with the maximum value of 85.13 g·kg$^{-1}$ by Luo et al. (2014)."

91. P13L7-11: Changed "and our results showed several plants (*Sanguisorba officinalis, Dendranthema indicum, Castanea henryi and Themeda japonica* ) and soil Eca content was a positive correlation, but most plant calcium content and soil ECa content was not relevant." into "Our results showed that most plants had no correlation relationship between soil ECa and plant Ca excepting several plant (*Sanguisorba officinalis, Dendranthema indicum, Castanea henryi and Themeda japonica* ) had a positive correlation between soil Eca and plant Ca content (Table. 4)."

92. P13L11: deleted "The Ca$^{2+}$ content in plant cells was proportional to soil Ca$^{2+}$."

93. P13L11-15: changed "Calcium-rich soils caused cells to absorb more calcium than the cells themselves require (White and Broadley, 2003). Zou et al. (2010) showed that soil ECa content and leaf calcium content are extremely significant positive correlation in pot experiment." into "But some study showed that Ca-rich soils caused cells to absorb more Ca than the cells themselves require (White and Broadley, 2003), and soil ECa content and leaf Ca content (Flue-cured Tobacco) had a significant positive correlation in pot experiment (Zou et al., 2010), which may be caused by species factors for the difference between our finds and their finds."

94. P13L15-23: inserted "The correlation between plant Ca content and soil ECa content reflects what extent soil Ca content influences plant Ca content, and may also reflect how different plants respond to differences in soil ECa content (Ji et al., 2009). The Ca content of *Sanguisorba officinalis* in the aboveground and belowground parts had a significant positive correlation (*p<0.01*) with soil ECa content, which indicated that *Sanguisorba officinalis* was affected greatly by soil ECa content. The Ca content of *Dendranthema indicum* (*p<0.05*) and *Castanea henryi* (*p<0.01*) in the belowground parts, showed a significant positive correlation (*p<0.01*) with soil ECa content, indicating that the belowground parts of these species were also greatly affected by soil ECa content. The Ca content of *Themeda japonica* in the aboveground parts showed a significant positive correlation (*p<0.01*) with soil ECa content, which indicated that the aboveground parts of *Themeda japonica* was also greatly affected by soil ECa content." Before "Two-way ANOVA of species and soil……"

95. P14L1-2: deleted two "-related"

96. P14L3: changed "in accordance with data reported previously" into "supported with data reported"

97. P14L5-8: changed "Since the transport of Ca was mainly one-way (upward), this result indicated that nitrogen-fixing plants were the most efficient in terms of the upward transport of Ca, and that Ca was mainly concentrated in the

aboveground parts of the plant; these findings were not consistent with those of Ji et al. (2009)." into "And this result indicated that nitrogen-fixing plants were the most efficient in the Ca upward transport, since the transport of Ca was mainly upward; which was not the same with those of Ji et al. (2009)."

98. P14L8: deleted "In their paper,"

99. P14L9-10: changed "," into ".", and add "They used three types of plants (pteridophytes, dicotyledons, monocotyledons) exclude nitrogen-fixing plants in their study, which may have a conflicting result compared with our current findings." after "the upward transport of Ca." after "……at the upward transport of Ca"

100. P14L10-18: deleted "and studied only three types of plants (pteridophytes, dicotyledons, monocotyledons) that did not include nitrogen-fixing plants, which may be the reason for the inconsistency of this previous data with our current findings. In terms of the Ca content of monocotyledons, we found significant differences (p<0.01) between the aboveground and underground parts, but the study by Ji et al. (2009) revealed that these differences were not significant. This may be because most of the monocotyledons collected were low-Ca plants.", and changed into "We found significant differences (p<0.01) between the aboveground and belowground parts in Ca content of monocotyledons in our study. However, Ji et al. (2009) revealed that no significant differences between the aboveground and belowground parts in Ca content of monocotyledons. This phenomenon may contribute the most of the monocotyledons sample plants were low-Ca plants."

101. P14L18-21: deleted "Owing to the fact that the aboveground parts of low-Ca plants maintain a lower Ca content for different grades of rocky desertification," and changed "a significant difference was found between the aboveground and underground parts in monocotyledons." into "A significant difference was found between the aboveground and belowground parts in monocotyledons, which may be because low-Ca plants maintain a lower Ca content in different grades of rocky desertification."

102. P15L2- P16L6: deleted "P15L2- P16L6 "content

103. P16L7-10: inserted "The different plant functional groups revealed the differences in Ca content (Fig. 3). In some cases, even within the same plant, there was an inconsistent correlation between Ca content in the aboveground and belowground parts and the soil ECa content. Collectively, these findings showed that not all plants adapted to soil high Ca environments in the same way, and that they exhibited a variety of adaptive mechanisms."

104. P16L12-16: inserted "(Grubb and Edwards, 1982). The capacity of these plants which was able to adapt to high Ca soil environments can be reflected by two indicators: (i) the correlation between Ca content in the aboveground parts of the plants and soil ECa content; (ii) the species differences in terms of the Ca content of the aboveground parts of plants. Thus, based on the above two indicators, the plants were classified into the following groups: Ca-indifferent plants, high-Ca plants and low-Ca plants (Ji et al., 2009)." After "……the Ca demand of the plant's physiological activity"

105. P16L16-19: deleted "The research conducted by Ji et al. (2009) was based on the differences in correlation between the Ca content of the aboveground parts of plants and its soil Ca content; these authors analyzed the capacity of plants to adapt to high Ca environments, and divided the dominated species into Ca-indifferent plants, high-Ca plants and low-Ca plants."

106. P16L21-P17L1: inserted "In both high-Ca and low-Ca soil environments, the Ca-indifferent plants can survival

normally. And the Ca content of them changes correspondingly with the change of soil ECa content. The physiological activities of high-Ca plants had a higher demand for Ca and may have a strong ability to enrich soil Ca. The physiological activities of low-Ca plants had a lower demand for Ca and could alleviate high Ca stress by inhibiting the absorption of Ca through the root system and its upward transport." after "……. vegetation restoration in rocky desertification areas."

107. P17L2-10: inserted "These results are of great significance to the vegetation restoration in karst areas. High-Ca plants should be preferentially selected (such as *Pyracantha fortuneana*, *Rhus chinensis*, and *Loropetalum chinense*, *Serissa japonica*), followed by Ca-indifferent plants (such as Sanguisorba officinalis, *Castanea henryi*, and *Dendranthema indicum*). Low-Ca plants also have a strong adaption ability on high calcium environments, and it can be used as an alternative species to increase species diversity during the process of ecological restoration. Our findings not only have important guiding significance for solving the problem of rocky desertification in China, but also provide species screening ideas for the rocky desertification ecosystem restoration in other parts of the world. Rocky desertification is a major ecological problem in karst areas. It is necessary to further explore other nutrient elements in soil during vegetation restoration, and long-term positioning observation is crucial for the study of this issue.

108. P17L11: changed "Conclusion" into "Conclusions"

109. P17L12-13: deleted "followed by" add "and" before "LRD"

110. P17L13-14: deleted the sentence" Significant differences were detected for both soil ECa and TCa content when compared between the rocky side and non-rocky side of each grade of rocky desertification areas."

111. P17L15-16: add "sites" after "studied", deleted "Furthermore", and changed "significant" into "Significant"

112. P17L17: deleted "(p<0.05)

113. L17L18: deleted "the" after "but had no significant effect on"

114. P17L19-20: put "," after "*Themeda japonica*", deleted "For these plants," and put "had"

115. P17L21: changed "High-Ca plants included *Pyracantha fortuneana*," into "High-Ca plants in our study were *Pyracantha fortuneana*,"

116. P17L22- P18L1: deleted "In this case", and changed "the aboveground parts of these plant were able to maintain a higher Ca content under conditions of variable soil ECa content." into "the aboveground parts of these plants were able to absorb a high Ca content from various of ECa content soils.

117. P19L23-24: deleted reference "Xiang, H., Zhang, L., and Chen, J.: Effects of calcium concentration in solution on calcium content in the seedlings of five fig plants, Guihaia, 23, 165–168, 2003. (in Chinese with English abstract)"

118. P20L6-8: add reference "Luo, X. Q., Wang, S. J., Zhang, G. L., Wang, C. Y., Yang, H. Y., and Liao, X. R.: Effects of calcium concentration on photosynthesis characteristics of two fern plants, Ecology and Environmental Science., 22, 258-262, 2013. (in Chinese with English abstract)"

119. P20L9: deleted reference "Yin, L. P.: Plant Nutrition Molecular Biology and Signal Transduction, Science Press, 2006. (in Chinese with English abstract)"

120. P21L2-3: add reference "Qi, Q. W., Hao, Z., Tao, J. J., and Kang, M.: Diversity of calcium speciation in leaves of *Primulina* species (Gesneriaceae), Biodiversity Science., 21, 715-722, 2013. (in Chinese with English abstract)"

121. P21L4-5: deleted reference "Li, Q. Y., Ge, H. B., Hu, S. M., and Wang, H. Y.: Effects of sodium and calcium salt stresses on strawberry photosynthesis, Acta Bot. Boreali-Occidental Sinica., 26, 1713–1717, 2006. (in Chinese with English abstract)"

122. P21L11: deleted reference "Simpson, J. F. H.: A chalk flora on the Lower Greensand: its use in interpreting the calcicole habit, J. Ecol., 26, 218–235, 1938."

123. P21L14-16: add reference "Wang, C. Y., Wang, S. J., Rong. L., and Luo, X. Q.: Analyzing about characteristics of calcium content and mechanisms of high calcium adaptation of common pteridophyte in Maolan karst area of China, Chinese J. Plant Ecol., 35, 1061-1069, 2011. (in Chinese with English abstract)"

124. P22L4-5: deleted reference "Feng, X. Y., Hu, Z. P., and Yi, Y.: Variation of proline and soluble protein content in leaves of Eurycorymbus cavalerieian and Pinus armandii under Ca2+ stress, Guizhou Agricultural Sciences., 38, 169–170, 2010. (in Chinese with English abstract)"

125. P22L17-18: deleted reference "Tu, Y. L.: Floristic and ecological characteristics in the karst shrub of Guizhou, Journal of Guizhou Normal University: Natural Science, 13, 1–8, 1995. (in Chinese with English abstract)"

126. P22L22-23: deleted reference "Zhou, Y. C.: A study on the part plants' main nutrient elements content of Guizhou karst region, Journal of Guizhou Agric., 16, 11–16, 1997. (in Chinese with English abstract)"

127. P24: added "Table 1. Basic description for different grades of rocky desertification sites"

128. P25: changed "Table 1" into 'Table 2", and change "point" into "location

129. P26: changed "Table 2" into 'Table 3"

130. P29L11-12: deleted "Annual herbs include annual and biennial herbs, while deciduous shrubs include deciduous trees with a height < 2 m or a ground diameter < 3 cm."

131. We changed all "underground" into "belowground" in the entire manuscript. And reworded all words "calcium" into "Ca" in the entire manuscript.

**Part 3:* marked-up manuscript version**

[revised manuscript text omitted]

---

## Author Response (ED1)

**Responses to the Comments**

**We highly appreciate the valuable comments and suggestions by the editor and anonymous reviewer on our manuscript. We have attempted to address each point raised by the editor and the reviewers. The following is our detail responses we have made, with reference to the order of the comments by the editor and the reviewers. We have asked an English editor from '_The Charlesworth Author Services Team (http://www.charlesworthauthorservices.com/)_' to improve the language. Hope this version will meet the requirements of BG.**

**Part 1:* Point-by-point response to Comments by the Editor (from attachment file)**

Note 1: suppress in

**Re.1 Thanks for the positive suggestion. Done!**

Note 2: This sentence is vague and seems a repetition of sentence of line 9. Could you explain more specifically the socioeconomic impact of this degradation

**Re.2 Thanks for your suggestion. We rewrote this sentence as: "Rocky desertification could lead to frequent natural disasters, reduce human survival and development space, threaten local people's production, life and life safety, cause ecological deterioration, reduce arable land resources, aggravate poverty, and affect sustainable economic and social development (Jing et al., 2016). In other words, Rocky desertification is an extreme form of land degradation in karst areas, and it has become a major social problem in terms of China's economic and social development (Sheng et al., 2015).".**

Note 3: what do you want to say here with "From the origin of desertification". It could be nice if you better explain the origin of this degradation. Is it due to an intensive cultivation of soils? or due to an urbanisation of this area? Climatic origin?

**Re.3 Sorry for the puzzle. We added new content to explain the origin of rock desertification degradation. They are "Given the origin of rocky desertification, the main factors that lead to rocky desertification are unreasonable human activities (reclamation on steep slope), causing damage to vegetation and exacerbating rocky desertification; its remediation should focus on vegetation restoration (Wang et al., 2004)."**

Note 4: this part of sentence can be more synthetic: Ca2+ is an essential nutrients for plant growth and

participate to....

**Re.4 Thanks. Done ! .**

Note 5: I do not understand the link between the different ideas of your sentence. I think you are talking about coupling between nutrient cycles, but this is not understandable. Develop your ideas

**Re.5 Sorry for this unclearly expression. We added content in order to make it easier to understand. They are "And $Ca^{2+}$ is a very important signal component in plants responsive to environmental stresses. $Ca^{2+}$ signal takes the influential role as a second messenger in hormone signal transduction, particularly in the abscisic acid signal transduction process (Hetherington, et al, 2004). Plants can adapt to high salt, drought and high temperature environments by activating the $Ca^{2+}$ signal transduction pathway (Bressan et al., 1998). $Ca^{2+}$ is also involved in nutrient cycling coupling process, for example, in the absence of nutrients (such as phosphorus), plants will inhibit the activity of nitrate reductase, thereby inhibiting the absorption of nitrate nitrogen, and ultimately inhibiting the absorption of $Ca^{2+}$ (Reuveni et al., 2000)."**

Note 6: we imagine that the association of pectin and Ca gives pectin-Ca...no need to write that. However, it might be important to specifiy which cell functions is controlled by this compound.

**Re.6 Thanks for your suggestion. We changed them as: "$Ca^{2+}$ combines with pectin in the cell walls of plants to form pectin Ca, which is a vital component of the intercellular layer in cell wall, and can buffer the compression between cells without hindering the expansion of cell surface area (Kinzel, 1989)."**

Note 7: suppress "the"

**Re.7 Thanks. Done. And this sentence was moved ahead.**

Note 8: This looks like the description of Ca function in cell physiology. This should maybe transferred in the previous paragraph. This reference to signal transduction is not enough supported by explanation. Why Ca is important for rignal transduction?

**Re.8 Thanks for the positive suggestion. Changed and moved to the "Role of $Ca^{2+}$ in plant physiology:" part.**

Note 9: it seems this is an important process. Please take time to present it

**Re.9 Yes. But moved to the paragraph of "Role of $Ca^{2+}$ in plant physiology:"**

Note 10: the release in (Ca goes into the vacuole) or from (Ca flows out)? Please take your time to well

explain

**Re.10 Yes. There are Ca²⁺ channels Ca²⁺ pump and Ca²⁺/H⁺ reverse conveyor on tonoplast. The former controls Ca²⁺ outflow, and the latter two pump cytoplasmic Ca²⁺ into vacuole (Wu, 2008).**

Note 11: There are not much informations in the different specific variability in plant Ca concentration and tolerance. Nothing is known about which species are tolerant? and by which mechanisms!? I do not believe that

**Re.11 Thanks for the positive suggestion. The following are information we added:**

**"Plants maintain low calcium content in aboveground part by reducing calcium uptake and transporting from underground part to aboveground part. This type of plant has *Nephrolepis auriculata, Parathelypteris glanduligera, Cyrtomium fortunei,Pteris vittata,* and so on. In contrast, other plants have a higher demand for calcium. For example, *Cayratia japonica* and *Corchoropsis tomentosa*. these plants maintain high calcium content by enhancing calcium uptake and transporting from underground to aboveground (Ji et al., 2009)."**

Note 12: But this title is a repetition of the same idea developped in the paragraph "Mechanism of plant defense to high soil Ca concentration". You should merge the redondant paragraphs (or find another way to present your ideas if you really think it's different ideas).

**Re.12 Thanks, very good suggestions! We merged them.**

Note 13: cristals in cristals? this seems weird

**Re.13 Sorry for the puzzling expression, and we have corrected it .**

Note 14: this is a repetition of the previous sentence, merge these two sentences

**Re.14 Thanks. Done.**

Note 15: exported where? Do you mean that plant concentrate ca in mature leave?

**Re.15 Sorry. Our means are: "When the leaves matured, excess Ca²⁺ in plants is excreted via stomata on the back of the leaves, thereby maintaining a lower concentration of leaf Ca (Musetti and Favali, 2003)."**

Note 16: what do you mean by growth habits? growth rate?

**Re.16 Sorry for the ambiguous word. It was growth conditions.**

Note 17: I am not sure to understand your use of word species. Are they different species between these two soils (calcareous and acidic soils). If yes why do you not insert the name of these species. Maybe

you mean different ecotypes of the same species?

**Re.17 Sorry for the confusing. And those sentences are:**

**"Qi et al. (2013) found that a significant difference in calcium content among *Primulina* species (*P. linearifolia, P. medica, P. swinglei, P. verecunda, P. obtusidentata, P. heterotricha,* and so on) from different soil types. and the average Ca content (2,285.6 mg/kg) in *Primulina* from calcareous soil was higher than the average calcium content of *Primulina* from both acid soil (1,379.3 mg/kg) and Danxia red soil (1,329.1 mg/kg)."**

Note 18: I do not really understand the message of your sentence: do you mean that differences among plant species in term of strategy to regulate Ca concentration in cell and capability to adapt to high soil Ca offer opportunity to identify the most relevant plants for the restoration of karst area? If yes you should better explain this idea.

**Re.18 I am sorry for the puzzling expression. This sentence should be**

**"There are variations in soil Ca content among different areas. And there are differences between calcareous and non-calcareous plants in terms of Ca absorption, transport, storage and other physiological processes. These differences need to be taken into account in order to identify the variety of plants able to adapt to high-Ca environments. However, to date, the mechanisms by which plants adapt to high Ca conditions, particularly in karst areas, and the Ca2+ dynamics of plants and soil are not well understood."**

**Part 2:* A list of all relevant changes made in the manuscript:**

**Abstract:**

1. P1L10: changed "Its" into "In these areas, the".
2. P1L11-12: changed "which" into "that", and deleted "in such rocky desertification areas".
3. P1L12-13: changed "Consequently, the screening of plant species, which can adapt to soil high $Ca^{2+}$ environments, is a critical step for vegetation restoration." into "Consequently, the screening of plant species that can adapt to high $Ca^{2+}$ soil environments is a critical step in vegetation restoration.".
4. P1L13-15: changed "In this study, three different grades of rocky desertification sample areas (LRD, light rocky desertification; MRD, moderate rocky desertification; IRD, intense rocky desertification) were selected in karst areas of southwestern Hunan, China." into "In this study, three grades of rocky desertification sample areas were selected in karst areas of southwestern Hunan, China (LRD: light rocky desertification; MRD: moderate rocky desertification; and IRD: intense rocky desertification).".
5. P1L16-17: changed "(1 in rocky side areas, 3 in non-rocky side areas)" into " (1 in rocky-side areas, 3 in non-rocky-side areas)".

6. P1L18: changed "exchange" into "exchangeable", changed "under" into "in".

7. P1L24-P2L3: changed "According to the differences in $Ca^{2+}$ content between the aboveground and belowground parts of 17 dominant species (important value , *IV*>1) and their correlations with soil ECa content, these 17 species can be divided into three categories: Ca-indifferent plants, high-Ca plants and low-Ca plants." into "Of the 41 plant species that were sampled, 17 were found to be dominant(important value >1). The differences in $Ca^{2+}$ content between the aboveground and belowground parts of the 17 dominant species were calculated, and their correlations with soil ECa content were analyzed. The results showed that these 17 species can be divided into three categories: Ca-indifferent plants, high-Ca plants and low-Ca plants.".

8. P2L3: changed "Our results" into "These findings"; changed "provides" into "provide".

**1 Introduction**

9. P2L7: changed "Karst is a kind of typical calcium (Ca)-rich environment and a unique ecological environment system" into "Karst is a calcium-rich environment and a unique ecological system"

10. P2L8: changed "world" into "world's".

11. P2L9-11: deleted "Rocky desertification is an extreme form of land degradation in karst areas, and it has become a major social problem in terms of China's economic and social development (Sheng et al., 2015).".

12. P2L11-12: Changed "The severity of rocky desertification was ranked in fourth in Hunan Province of China" into "The Hunan Province of China has been ranked fourth for the severity of rocky desertification".

13. P2L12-16: add this sentence "Rocky desertification could lead to frequent natural disasters, reduce human survival and development space, threaten local people's production, life and life safety, cause ecological deterioration, reduce arable land resources, aggravate poverty, and affect sustainable economic and social development", and deleted "The restoration and reconstruction of rocky desertification ecosystems has become the urgent environment improvements, regional economic development by using agroforestry system and helping to support people out of poverty"

14. P2L17-18: added "In other words, Rocky desertification is an extreme form of land degradation in karst areas, and it has become a major social problem in terms of China's economic and social development (Sheng et al., 2015).", after "(Jing et al., 2016).".

15. P2L18: Changed "high calcium Ca" into "high calcium (Ca)".

16. P2L20: change "From" into "Given".

17. P2L20-22: add "the main factors that lead to rocky desertification are unreasonable human activities (reclamation on steep slope), causing damage to vegetation and exacerbating rocky desertification;" after "desertification,".

18. P2L23: changed "which" into "that".

19. P3L2-4: added "Over recent decades, progress has been made in identifying the cellular compartments (e.g., endoplasmic reticulum, chloroplasts and mitochondria) that regulate Ca balance and signal transduction in plants (Müller et al., 2015)." After "in plant physiology.".

20. P3L4-5: deleted "$Ca^{2+}$ is one of the most essential nutrients needed for the regulation of plant growth and is also plant signal sensor (second messenger) under conditions of environment stress"; and add "$Ca^{2+}$ is an essential nutrients for plant growth and also participate to a transduction" before "(Poovaiah and Reddy, 1993; Hepler, 2005; Hong-Bo and Ming, 2008; Batistič and Kudla, 2012)".

21. P3L6-10: added "And $Ca^{2+}$ is a very important signal component in plants responsive to

environmental stresses. $Ca^{2+}$ signal takes the influential role as a second messenger in hormone signal transduction, particularly in the abscisic acid signal transduction process (Hetherington, et al, 2004). Plants can adapt to high salt, drought and high temperature environments by activating the $Ca^{2+}$ signal transduction pathway (Bressan et al., 1998). $Ca^{2+}$ is also involved in nutrient cycling coupling process, for example," after "(Poovaiah and Reddy, 1993; Hepler, 2005; Hong-Bo and Ming, 2008; Batistič and Kudla, 2012)."

22. P3L10: "changed ". In' into "in".

23. P3L13-14: Added "the intercellular layer in cell wall, and can buffer the compression between cells without hindering the expansion of cell surface area" after "of"; deleted "the cell wall".

24. P3L17-19: deleted "Over the past decades, the progress has been made in identifying the cellular compartments (e.g., endoplasmic reticulum, chloroplasts and mitochondria) that regulate Ca balance and signal transduction in plants (Müller et al., 2015).".

25. P3L20-22: deleted "Plants can be adapted to high salt, drought and high temperature environments by activating the Ca2+ signal transduction pathway (Bressan et al., 1998).".

26. P4L2-12: added ". Some plants fix excess Ca2+ by forming calcified deposits in root tissue in order to limit the upward transport of Ca2+ (Musetti and Favali, 2003). In addition, Ca oxalate crystals in the plant cells play a role in regulating plant Ca content (Ilarslan et al., 2001; Pennisi and McConnell, 2001; Volk et al., 2002), and some plants will form Ca oxalate crystal cells in order to fix excess Ca2+ (Moore et al., 2002). Furthermore, an active Ca efflux system plays an important role in the adaptation of plants to high-Ca environments (Bose et al., 2011). For example, when the leaves matured, excess Ca2+ in plants is excreted via stomata on the back of the leaves, thereby maintaining a lower concentration of leaf Ca (Musetti and Favali, 2003). The regulation of internal Ca storage depends predominantly on plasma membrane Ca transport and intracellular Ca storage; collectively these processes can regulate the intracellular Ca2+ concentration to a lower level (Bowler and Fluhr, 2000). Plants that adapt to high-Ca environments promote excess Ca2+ flow through the cytoplasm or store Ca2+ in vacuoles via the cytoplasmic Ca2+ outflow and influx system (Shang et al., 2003; Hetherington and Brownlee, 2004; Wang et al., 2006)." after "means of Ca storage (Ranjev et al., 1993).".

27. P4L14-16: Added "There are $Ca^{2+}$ channels Ca2+ pump and $Ca^{2+}/H^+$ reverse conveyor on tonoplast. The former controls $Ca^{2+}$ outflow, and the latter two pump cytoplasmic $Ca^{2+}$ into vacuole (Wu, 2008)." After "……the release of $Ca^{2+}$ in vacuoles (Peiter, 2011)."

28. P4L18: added "the" before "cytoplasm combines".

29. P4L19: added "the" before "phosphorus metabolism".

30. P4L20-P5L1: added "Plants maintain low calcium content in aboveground part by reducing calcium uptake and transporting from underground part to aboveground part. This type of plant has *ephrolepis auriculat*, *Parathelypteris glanduligera*, *Cyrtomium fortunei*,*Pteris vittata*,and so on. In contrast, other plants have a higher demand for calcium. For example, *Cayratia japonica* and *Corchoropsis tomentosa*. these plants maintain high calcium content by enhacing calcium uptake and transporting from underground to aboveground." after "…. growth (White and Broadley, 2003; Hirschi, 2004).".

31. P5L2-12: deleted "Plants adaptation to high Ca soil environment: Some plants fix excess $Ca^{2+}$ by forming calcified deposits in root tissue in order to limit the upward transport of $Ca^{2+}$ (Musetti and Favali, 2003). In addition, Ca oxalate crystals in the plant's crystal cells play a role in regulating plant Ca content (Ilarslan et al., 2001; Pennisi and McConnell, 2001; Volk et al., 2002). In a high-

Ca environments, some plants will form Ca oxalate crystal cells in order to fix excess $Ca^{2+}$ (Moore et al., 2002). Furthermore, an active Ca efflux system plays an important role in the adaptation of plants to high- Ca environments (Bose et al., 2011): Excess $Ca^{2+}$ in plants is exported from mature leaves to the outside, thereby maintaining a lower concentration of leaf Ca (Musetti and Favali, 2003). The regulation of internal Ca storage depends predominantly depends on plasma membrane Ca transport and intracellular Ca storage; collectively these processes can regulate the intracellular $Ca^{2+}$ concentration to a lower level (Bowler and Fluhr, 2000). Plants that adapt to high-Ca environments promote excess $Ca^{2+}$ flow through the cytoplasm or store $Ca^{2+}$ in vacuoles via the cytoplasmic $Ca^{2+}$ outflow and influx system (Shang et al., 2003; Hetherington and Brownlee, 2004; Wang et al., 2006).".

32. P5L13: changed "ECa" into "exchangeable $Ca^{2+}$ (ECa)".

33. P5L15: changed "TCa" into "total $Ca^{2+}$ (ECa)".

34. P5L15: changed "area were" into "areas was".

35. P5L16: changed "habits" into "conditions".

36. P5L20-24: deleted "that". Changed "Qi et al. (2013) found that a significant difference in calcium content among *Primulina* species from different soil types, with high average calcium content (2,285.6 mg/kg) in *Primulina* from calcareous soil relative to low levels present in was *Primulina* from both acid soil (1,379.3 mg/kg) and Danxia red soil (1,329.1 mg/kg)." into "Qi et al. (2013) found that a significant difference in calcium content among *Primulina* species (*P. linearifolia, P. medica, P. swinglei, P. verecunda, P. obtusidentata, P. heterotricha,* and so on) from different soil types. and the average Ca content (2,285.6 mg/kg) in *Primulina* from calcareous soil was higher than the average calcium content of *Primulina* from both acid soil (1,379.3 mg/kg) and Danxia red soil (1,329.1 mg/kg).".

37. P5L24-P6L4: changed "There are variations in soil Ca content among different areas and differences between calcareous and non-calcareous plants in terms of Ca absorption, transport and storage and other physiological processes. These differences need to identify the variety of the plants to adapt with high Ca environments. However, to data, the mechanisms by which plants adapt to high Ca conditions, particularly in karst areas and the $Ca^{2+}$ dynamics of plants and soil are not well understood." into 'There are variations in soil Ca content among different areas. And there are differences between calcareous and non-calcareous plants in terms of Ca absorption, transport, storage and other physiological processes. These differences need to be taken into account in order to identify the variety of plants able to adapt to high-Ca environments. However, to date, the mechanisms by which plants adapt to high Ca conditions, particularly in karst areas, and the $Ca^{2+}$ dynamics of plants and soil are not well understood.".

38. P6L8: changed "rocky and non-rocky side areas" into "rocky-side and non-rocky-side areas"

**2 Materials and methods**

39. P6L13: changed "is" into "was". And deleted "of" added "in".

40. P6L14: deleted "; see Fig. 1", and added ", as shown in Fig.1" after bracket ")".

41. P6L20: changed "expose" into "exposure".

42. P6L21: added ". These index" before "were quantified".

43. P7L1-2: deleted "LRD,", and added "(LRD)" after "light rocky desertification".
    deleted "MRD,", and added "(MRD)" after "moderate rocky desertification".
    changed "LRD," into "and", and added "(IRD)" after "intense rocky desertification".

44. P7L3: changed "," into "which", changed "including" into "included".

45. P7L4: changed "The sample collection of samples in these three sample areas were conducted in October 2016." into "The collection of samples in these three sample areas was conducted in October 2016.".

46. P7L11: changed "we measured the TCa and ECa relating to the quadrat soil" into "we measured the soil TCa and ECa content of each quadrat".

47. P7L18: changed "The biennial herbs were gathered to the 'Annual herbs'. The deciduous" into "Biennial herbs were gathered to the 'annual herbs'. Deciduous".

48. P7L20-21: added "analysis of variance" after "two-way".

49. P7L21: Deleted "these"; and added "widespread" after "17".

50. P7L22: deleted "analysis of variance"; and changed "(ANOVA)" into "ANOVA".

**3 Results**

51. P8L6: changed "that the" into "to be".

52. P8L7: changed "followed by" into "then".

53. P8L8-10: changed "Regarding the availability of Ca, the average Ca content was 59.75%, the MRD showing the highest content at 72.55%, followed by IRD at 58.98%, and LRD showing the lowest content at 47.72 % (Table. 2)." into "Regarding the availability of Ca, the average availability of Ca was 59.75%, with the MRD showing the highest value at 72.55%, followed by IRD at 58.98%, and LRD showing the lowest value at 47.72 % (Table. 2)."

54. P8L13: changed "Total of" into "A total of".

55. P8L14: deleted "Compare to t"; and added "T".

56. P8L17: deleted "of the plants were", and added "s showed".

57. P9L4: deleted "than" into "compared to".

58. P9L17: Changed "Within total 41 common plants species" into "Within the total of 41 common plant species".

59. P9L19: added "differences of" after "Data showed that the".

60. P9L21-22: changed "The Ca content in the belowground parts were highly significant difference not only among the species, and it throughout all the grades of rocky desertification (p<0.01)." into "Differences in the Ca content of the belowground parts were highly significant not only among species, but throughout all the grades of rocky desertification ($p<0.01$).".

61. P10L2: deleted "T", and added "Of t".

62. P10L6: deleted "T", added "With regard to t"; and changed "of" into "in".

63. P10L8: deleted "soil'; and added "soil" before "environments".

64. P10L14-15: Changed "These plants did not strictly control the absorption and transport of Ca and may be insensitive to the changes of their own Ca content, and their growth was less affected by soil Ca content." into "These plants did not exercise a strict control over the absorption and transport of Ca and may be insensitive to changes in their own Ca content. Moreover, their growth was less affected by soil Ca content.".

65. P10L15-17: changed "In addition," into "As", and added "the" after "for"; deleted ", then T", and added "T", and added "then" after "were".

**4 Discussion**

66. P11L3: added "also" after "the Ca content of soil". And changed "indicated" into "indicates".

67. P11L11-13: changed "affecting" into "for the different of";
    deleted "mean", and added "ed that the mean" after "Our research show";
    changed "of" into "in"; changed "expose" into "exposure".

68. P11L16: changed "But" into "However"; added "the" before "aboveground parts of plants (19.67 g·kg⁻¹)".

69. P11L21: added "detected" after "value of 85.13 g·kg⁻¹".

70. P11L21-P12L1: Changed "To most plant Ca content, the aboveground part was larger than the belowground part, and for a few plants Ca content, the aboveground part was lower than the belowground part (such as *Sanguisorba officinalis*, *Pyracantha fortuneana* and *Castanea henryi*), which was consistent with the findings of Wang et al. (2014)." into "For most plants, the Ca content in the aboveground part was higher than in the belowground part, but for a few plants the Ca content in the aboveground part was lower than in the belowground part (such as *Sanguisorba officinalis*, *Pyracantha fortuneana* and *Castanea henryi*), which was consistent with the findings of Wang et al. (2014).

71. P12L4: changed "had" into "which showed".

72. P12L6: changed ", and" into ". Additionally,".

73. P12L7-8: Deleted "for the difference between our finds and their finds" and ", which", and added ". The difference between the findings of these studies and ours" before "may be caused by species factors".

74. P12L13: added "also" before "showed a significant positive".

75. P12L20: changed ", which was supported with data reported by Ji et al. (2009)." into ". This finding is supported by data reported by Ji et al. (2009)."

76. P12L21-P13L3: changed ". And this result indicated that nitrogen-fixing plants were the most efficient in the Ca upward transport, since the transport of Ca was mainly upward; which was not the same with those of Ji et al. (2009). Ji et al. (2009) revealed that dicotyledons were the most efficient at the upward transport of Ca. They used three types of plants (pteridophytes, dicotyledons, monocotyledons) exclude nitrogen-fixing plants in their study, which may have a conflicting result compared with our current findings." into ", and this result indicates that nitrogen-fixing plants were the most efficient in Ca upward transport. In contrast, Ji et al. (2009) found that dicotyledons were the most efficient in the upward transport of Ca. They used only three types of plants (pteridophytes, dicotyledons, and monocotyledons) without researching nitrogen-fixing plants in their study, which may have produced a conflicting result compared with our current findings.".

77. P13L4: added "the" before "Ca content of monocotyledons in our study.".

78. P13L4: added "there are" after "Ji et al. (2009) revealed that".

79. P13L6: changed " may contribute" into "could be due to".

80. P13L16: changed "could be due to" into "may contribute".

81. P13L9: deleted "individual".

82. P13L14-15: changed "these findings showed that not all plants adapted to soil high Ca environments in the same way, and that they exhibited a variety of adaptive mechanisms." into "these findings show that not all plants adapted to high Ca soil environments in the same way, but rather exhibited a variety of adaptive mechanisms.".

83. P13L17-18: changed "which was" into "that are".

84. P13L20-21: changed " Thus, based on the above two indicators, the plants were classified into the following groups: Ca-indifferent plants, high-Ca plants, and low-Ca plants (Ji et al., 2009)." into "Thus, based on these two indicators, plants can be placed into the following groups: Ca-indifferent plants, high-Ca plants, and low-Ca plants (Ji et al., 2009).".

85. P13L22: Changed "our plants species" into "the 17 plant species"; changed ", which" into "that".

86. P14L1-2: Changed "survival" into "survive".
    Changed ". And the Ca content of them changes correspondingly with the change of soil ECa content." into ", and their Ca content changes correspondingly with changes in soil ECa content."

87. P145-L6: Deleted "selected", and added "selected" before "preferentially".

88. P14L7-8: changed "Low-Ca plants also have a strong adaption ability on high calcium environments," into "Low-Ca plants also have a strong ability to adapt to high calcium environments,".

89. P14L7-8: Changed "Low-Ca plants also have a strong adaption ability on high calcium environments," "Low-Ca plants also have a strong ability to adapt on high calcium environments,". And deleted "it".

90. P14L9-10: deleted "guiding", and changed "solving" into "guiding solutions to".

91. P14L11: added "ecosystem restoration in" after "but also provide species screening ideas for", deleted "the", changed "ecosystem restoration" into "areas".

92. P14L12: added ", and further explorations are required to solve this problem." after "…a major ecological problem in karst areas".

93. P14L14: changed "the study of" into "understanding".

**5 Conclusions**

94. P14L16: changed "indicated" into "indicate".

95. P14L16: changed "and" into "then".

96. P14L18: deleted "studied sites", and added "sites in our study ".

97. P14L19: added "plants in" before "each plant functional group.".

98. P14L20: Deleted "," and "had", and added "the" before "Ca content of the aboveground parts."

99. P14L21: Added ", which" after "Ca-indifferent plants".

100. P14L22: changed "had" into "showed", and deleted "existed".

101. P15L1-2: changed "The aboveground parts of these plants were able to absorb a high Ca content from various of ECa content soils." into "The aboveground parts of these plants were able to absorb a lot of Ca from soils with varying ECa content.".

102. P15L3-4: Changed "The aboveground parts of low-Ca plants were able to maintain a lower Ca content under conditions of variable soil ECa content." into "The aboveground parts of low-Ca plants were able to maintain a lower Ca content from soils with varying ECa content.".

103. Table 1, changed "expose" into "exposure", and changed "rarely grazing" into "rarely grazed"

104. Table 2, deleted "D", added "The d".

105. Table 3, deleted all "(IV), and changed "are' into "is".

106. References were adjusted in order.

[revised manuscript text omitted]

---

## Author Response (AR4)

**Responses to the Comments**

**We highly appreciate the valuable comments and suggestions by the editor on our manuscript. We have attempted to address each point raised by the editor. The following is our detail responses with reference to the order of the comments.**

5 ***Part 1:*** **Point-by-point response to Comments by the Editor (from attachment file)**

Note 1: I do not understand what do you mean by "important value"

**Re.1 Importance value is a comprehensive quantitative index of the status and function of species in community. In this paper, the important values of the plant species in sample plots were calculated separately for shrubs and herbs. The calculation formula is: Important value = (Relative Density + Relative Height + Relative Coverage)/3. Important value**

10 **is a term in plant community and forest science researches. And we added this formula in part "2.3 Data analysis".**

Note 2: This added sentence cannot intercalate like that. Write a specific sentence and developp a bit the unreasonable activities.

**Re.2 Thanks! This sentence is unaccustomed in the context. And we changed it as:**

**Given the origin of rocky desertification, the main factors that lead to rocky desertification are unreasonable human**

15 **activities. For example, the cultivation of crops on steep slope can cause vegetation destruction, soil erosion, and then rocky desertification. We should focus on vegetation restoration for the rocky desertification remediation (Wang et al., 2004).**

Note 3: start your sentence with lowercase.

20 **Re.3 Thanks. Done!**

Note 4: The next 8 lines do not deal with specific variability in plant $Ca^{2+}$. Or I do not understand exactly what do you mean by specific variability. For me it refers to difference between plant species. If it is the case, you must start with this aspect and

continue the description of mechanismes developped by these plant species to tolerate high $Ca^{2+}$.

**Re.4 Sorry for the confusing. In this paragraph we are talking about "The mechanism of the variation in plant Ca$^{2+}$ content in different species". So they were changed as:**

**"Variation of Ca$^{2+}$ content in species and soil", and those paragraphs have been integrated.**

Note 5: This sentence should rather be written like that "The soil Ca content increased with the grade of rocky desertification".

**Re.5 Thanks. Done!**

Note 6: These 2 sentences must be merged.

10 **Re.6 Thank you. Done!**

Note 7: you mean "in the tree studied sites?"

**Re.7 Sorry for the confusing! It was changed as:**

**"The mean soil ECa content was 1.46 g·kg$^{-1}$ in these three different grades of rocky desertification"**

Note 8: Is this comparison relevant? What this tobacco-planting soil is good reference?

**Re.8 Because they are all in the same place (Hunan province). We think that there is a certain degree of comparability on this point. In addition, referring to the study of ECa in Hunan, the focus is not on tobacco. It is in order to compare the difference in the average ECa content between rock desertification areas and non-rocky desertification areas.**

Note 9: OK why not, but did you make this comparison? Did they also study degraded, desertified, land?

**Re.9 Yes. They studied four soil types (basaltic conglomerates, Bohio; Andesite; volcanoclastic sediments with basaltic agglomerates, Caimito volcanic; foraminiferal limestone, Caimito marine) on Barro Colorado Island (BCI). But did not study land degradation. We also studied limestone soil types too, which are comparable.**

Note 10: thickness? I am the feeling you compare different soil layers here non? Maybe you meant "soil depth".

**Re.10 Thanks for the positive suggestion. It was changed as "soil organic layer depth".**

Note 11: This assertion is not really correct. The ratio can be maintained with different Ca concentrations in aboveground and

5 belowground biomass.

**Re.11 Sorry for the confusing. We changed them as :**

**There were no significant differences in plant Ca content among the different grades of rocky desertification either for the aboveground or belowground parts (p>0.05), indicating that the grade of rocky desertification had no obvious effect on the Ca content of the aboveground and belowground parts of the plants.**

Note 12: Why but? I do not see contradiction in what we wrote.

**Re.12 Thanks! 'But' was deleted.**

Note 13: some means plurial, some studies.

15 **Re.13 Thanks. Done!**

Note 14: Add "However"

**Re.14 Thanks. Done!**

20 Note 15: replace without researching by omitting.

**Re.15 Thanks. Done!**

Note 16: Maybe you can delete the sentence part after the comma. We understand.

**Re.16 Thanks. Done!**

Note 17: I do not understand what do you try to say. You mean you've a larger sample size of monocotyledons than Ji? And that the sample size of Ji was not sufficient to detect significant differences?

**Re.17 Sorry for the confusing sentence. The difference between our study and Ji maybe the sample size and different species. But after serious discussion, we found the main reason should be "different species". So we've corrected them as:**

**"However, Ji et al. (2009) revealed that there were no significant differences between the aboveground and belowground parts in the Ca content of monocotyledons. The main reason for this difference may be the different species. In addition, the Ca content of monocotyledons was lower than that reported for monocotyledons by Ji et al. (2009), highlighting the large difference in ability to absorb soil Ca among monocotyledon species."**

Note 18: replace by "highlighting the large difference in ability to absorb soil Ca among monocotyledon species" if I understand well your idea!

**Re.18 Thanks. Done!**

15 Note 19: some plant? Could you specify? plant species? plant types?

**Re.19 Sorry. It should be "the same plant species".**

Note 20: delete "In the present paper" and start directly with your idea.

**Re.20 Thanks. Done!**

Note 21: I do not understand this idea. Ca comes from the soil not the plant.

**Re.21 Sorry for this mistake. The "enrich" was changed by "absorb ".**

Note 22: This sentence is a bit strange. At this end we have the feeling that all plant species are usefull and that this work

cannot be used to phocus on some relevant plant species. Moreover, I will find important to underline the relevance of legumes on resotration of these areas: their tolerance to Ca is not the only interesting aspect. They also inject organic N into the system promoting soil organic matter accumulation. This accumulation will improve soil fertility, soil stability, soil water availability. Therefore, legumes offer a bouquet of services that might be crucial for the restoration of these surfaces

5    **Re.22 Thanks! Legumes are important, but not all legumes are tolerant to high calcium environments. In our study, species tolerance is a prerequisite, and the selected species is classified based on this precondition. These plants can normally grow in different grades of rocky desertification. These species are all useful for the recovery of rocky desertification areas, but there are difference in the calcium content they absorb from soil.**

**The original sentence has also been modified to:**

10   **"Low-Ca plants should only be used as an alternative species to increase species diversity during the process of ecological restoration".**

Note 23: From applied perspectives, is it more interesting to have plants that accumulate in aboveground part than belowground? Is it relevant to imagine that plant aboveground is harvested to export the excess of Ca present in these systems?

15   **Re.23 Yes. There are application prospects. Maybe we should focus on it in the future research. In this study, our main aims are species screening and their classification.**

**Part 2:* A list of all relevant changes made in the manuscript:**

20   1.  P2L16-17: Changed "Given the origin of rocky desertification, the main factors that lead to rocky desertification are unreasonable human activities (reclamation on steep slope), causing damage to vegetation and exacerbating rocky desertification.; its remediation should focus on vegetation restoration (Wang et al., 2004)." into "Given the origin of rocky desertification, the main factors that lead to rocky desertification are unreasonable human activities. For example,

the cultivation of crops on steep slope can cause vegetation destruction, soil erosion, and then rocky desertification. We should focus on vegetation restoration for the rocky desertification remediation (Wang et al., 2004).".

2. P2L19: changed "Over" into "over".

3. P4L2-3: deleted "Specific variability in plant $Ca^{2+}$ content and tolerance: The concentration of free $Ca^{2+}$ in vacuoles varies with plant species, cell type and environment, which may also affect the release of $Ca^{2+}$ in vacuoles (Peiter, 2011).".

4. P4L10-11: added "Variation of $Ca^{2+}$ content in species and soil: The concentration of free $Ca^{2+}$ in vacuoles varies with plant species, cell type and environment, which may also affect the release of $Ca^{2+}$ in vacuoles (Peiter, 2011)." before "Some species maintain low……".

5. P4L11: changed "plants" into "Some species".

6. P4L15-16: added respectively "part" after "underground" and "aboveground".

7. P4L17-19: moved "The mean soil exchangeable Ca (ECa) was 3.61 $g \cdot kg^{-1}$ in the Puding, Huajing, Libo and Luodian Counties of Guizhou Province, which is several times that of non-limestone areas in China (Ji et al., 2009). Wang et al. (2011) found that plant rhizosphere soil total Ca (TCa) content in calcareous soil areas was above 14.0 $mg \cdot g^{-1}$." to the back of "…...soil (1,379.3 mg/kg) and Danxia red soil (1,329.1 mg/kg)" in P5L3-6.

8. P5L7: changed "And" into "and".

9. P6L1: changed "lime" into "limestone".

10. P7L4-6: added "We calculated the important values (IV) by the following formula:
$$IV = \frac{\text{Relative Density} + \text{Relative Height} + \text{Relative Coverage}}{3}$$
" after "while the belowground part only included roots.".

11. P10L12-13: deleted "With the grades of rocky desertification increased, the Ca content of soil also increased. This", and added "The soil Ca content increased with the grade of rocky desertification, which".

12. P10L14-15: changed "in three rocky desertification areas" into "in these three different grades of rocky desertification"; added "province" after "Hunan".

13. P10L20: changed "organic layer thickness" into "soil organic layer depth".

14. P11L2-5: changed "There was no significant difference in plant Ca content between aboveground or belowground parts (p>0.05) across the different grades of rocky desertification. This indicates that the grade of rocky desertification had no obvious effect on the Ca content of the aboveground and belowground parts of the plants studied herein." into "There were no significant differences in plant Ca content among the different grades of rocky desertification either for the aboveground or belowground parts ($p$>0.05), indicating that the grade of rocky desertification had no obvious effect on the Ca content of the aboveground and belowground parts of the plants.".

15. P11L16: changed " But some study" into "Some studies".

16. P12L10: changed "They" into "However, they".

17. P12L11-12: changed "without researching" into "omitting". And deleted ", which may have produced a conflicting result compared with our current findings".

18. P12L14-18: changed "This phenomenon could be due to the most of the monocotyledons sample plants were low-Ca plants. In our study, a significant difference was found between the aboveground and belowground parts in monocotyledons, which may be because low-Ca plants maintain a lower Ca content in different grades of rocky desertification." into "The main reason for this difference may be the sample size and the species differences".

19. P12L19-20: changed "indicating that different monocotyledons showed differing abilities to absorb soil Ca." into "highlighting the large difference in ability to absorb soil Ca among monocotyledon species".

20. P12L22-23: added "species" after "even within the same plant". Changed "was" into "were".

21. P13L8: deleted "In the present paper,", and changed "we" into "We".

22. P13L12: changed "enrich" into "absorb".

23. P13L17-18: deleted "also have a strong ability to adapt to high calcium environments, and they can" and added "should only".

24. P14L3: deleted "for", and added "of plants in".

25. P14L4: added "plant" before "Ca content were found….".

26. P14L5: deleted "of plants".

**Part 3: marked-up manuscript version**

[revised manuscript text omitted]